# Strengthen Out-of-Distribution Detection with Uncertainty-Driven Adaptively Rectified Backpropagation

## Abstract

Out-of-distribution (OOD) detection aims to ensure AI system reliability by detecting inputs outside the training distribution. Recent work shows that overfitting during later stages of training can hurt OOD detection. To overcome overfitting, several methods attempt to distill the model after training or prune the model during training from a model-centric perspective. In contrast, this paper proposes a data-centric end-to-end solution called Uncertainty-driven Adaptively Rectified Backpropagation (UARB), which follows the principle that once the model has mastered an instance, training on it should stop to prevent overfitting. UARB considers an instance mastered if the zero-order and second-order differences of its uncertainty value remain within a small range around zero, offering a more consistent measure of an instance's learning status. Additionally, since different classes exhibit varying optimization progress, using a fixed threshold to determine when to exclude an instance from backpropagation is theoretically unsound. UARB develops an adaptive threshold by incorporating class-informed statistics to determine when to exclude an instance. Extensive experiments demonstrate that UARB can enhance OOD detection performance.

## 1 Introduction

In real-world open-environment applications, machine learning models inevitably encounter samples from unknown categories that differ markedly from the training data distribution (Boyd et al., 2023). These out-of-distribution (OOD) samples pose significant risks to model reliability, particularly in critical fields such as medical diagnostics (Nair et al., 2020) and network intrusion detection (Corsini & Yang, 2023). Assigning OOD samples to known categories can result in severe consequences. Studies reveal that deep neural networks tend to produce overconfident, erroneous predictions for OOD inputs, highlighting a systemic vulnerability that compromises AI system safety. Consequently, robust OOD detection mechanisms are vital for ensuring safe and reliable model deployment in open-world scenarios. By effectively detecting and rejecting OOD samples, we can increase the decision-making credibility of AI systems, providing essential safeguards in high-stakes domains (Jiang et al., 2024).

Recent years have witnessed significant advancements in OOD detection. Early approaches primarily focused on developing effective scoring functions to quantify uncertainty, including Maximum Softmax Probability (MSP) (Hendrycks & Gimpel, 2016b), ODIN (Liang et al., 2017), and energy-based scores (Liu et al., 2020). These methods aim to create distinct score distributions between in-distribution (ID) and OOD data. Subsequent research introduced more sophisticated techniques such as Mahalanobis distance (Lee et al., 2018) and Virtual Logit Matching (ViM) (Wang et al., 2022), which demonstrated improved discriminative capabilities. Beyond post-hoc scoring methods, training-time regularization has emerged as a promising direction. One line of work explores enhanced training strategies to learn more discriminative representations (DeVries & Taylor, 2018; Sehwag et al., 2021; Wei et al., 2022b). Another key approach involves exposing models to auxiliary outlier data during training, as seen in Outlier Exposure (OE) (Hendrycks et al., 2018). Although these approaches show considerable success, limited attention is given to the models' intrinsic OOD detection capabilities, which constrains their full potential.

To provide an intuitive explanation of this issue, similar to (Yang & Xu, 2025), we monitor the model's performance on ID classification and OOD detection throughout the entire training process, as shown in Figure 1. The results show that OOD detection performance initially improves and then declines. According to (Cheng et al., 2025), the underlying cause of this phenomenon can be attributed to overfitting on mastered instances during the later stages of training. (Zhu et al., 2023) confirmed that this phenomenon is prevalent across a variety of training settings. In response, from the model's perspective, (Cheng et al., 2025) adopted pruning to eliminate overfitting by avoiding redundant features, and (Yang & Xu, 2025) adopted progressive self-knowledge distillation. While these techniques can superficially mitigate overfitting, they alone cannot resolve the data-level issues causing overfitting and rely on an additional validation set. The most relevant to our work is UM (Zhu et al., 2023), which determines whether a sample is overfitted based on its loss, and then "forgets" such samples by increasing their loss. However, relying solely on the loss itself is insufficient to effectively identify overfitted samples, and reversing gradient optimization for such samples can easily lead to unstable model training. Furthermore, these approaches overlook the issue of inconsistent learning paces across different classes. These issues ultimately hinder effective enhancement of OOD detection capability.

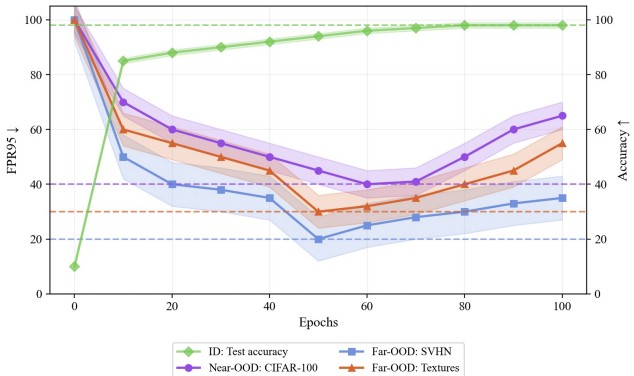

Figure 1: The FPR95 (lower values indicate better OOD detection) curves for various OOD datasets and the ID test accuracy curve, when the ID dataset is CIFAR-10.

To address the above issues, we propose a data-centric end-to-end solution called Uncertainty-driven Adaptively Rectified Backpropagation (UARB), which follows the principle that once the model has mastered an instance, training on it should stop to prevent overfitting. UARB considers an instance mastered if the zero-order and second-order differences of its uncertainty value remain within a small range around zero, offering a more consistent measure of an instance's learning status. Once an instance is mastered, it is excluded from further backpropagation. Additionally, since different classes exhibit varying optimization progress, using a fixed threshold to determine when to exclude an instance from backpropagation is theoretically unsound. Accordingly, UARB develops an adaptive threshold by incorporating class-informed statistics to determine when to exclude an instance. UARB effectively mitigates overfitting through instance-level early stopping with an adaptive threshold. Extensive experiments on benchmarks demonstrate that the UARB method can enhance OOD detection performance. Our main contributions are as follows:

- We propose a data-centric end-to-end solution called Uncertainty-driven Adaptively Rectified Backpropagation (UARB) to mitigate overfitting in OOD detection.
- We exclude mastered instances from further backpropagation according to the zero-order and second-order differences of each instance's uncertainty value.
- We develop an adaptive threshold by incorporating class-informed statistics to determine when to exclude an instance from backpropagation.
- Extensive experiments on benchmarks demonstrate that the UARB method enhances OOD detection performance.

## 2 METHODOLOGY

In this paper, we propose a data-centric end-to-end solution called Uncertainty-driven Adaptively Rectified Backpropagation (UARB) to mitigate overfitting in OOD detection. First, we introduce a principled criterion for identifying instances that have been mastered by the model (Section 2.2). Second, we leverage this criterion to implement instance-level early stopping, excluding mastered instances from further backpropagation using a fixed threshold (Section 2.3). Third, we develop an

adaptive threshold by incorporating class-informed statistics, enabling instance-level early stopping tailored to class-specific learning dynamics (Section 2.4).

## 2.1 LEARNING SET-UP

We consider a training set $\mathcal{D}_{in}^{train} = \{(\boldsymbol{x}_i, y_i)\}_{i=1}^n$ drawn i.i.d. from distribution $P(\mathcal{X}, \mathcal{Y})$, where $\mathcal{X} = \mathbb{R}^d$ is the input space, $\mathcal{Y}$ is the label space, $\mathcal{Y} = \{1, 2, \ldots, K\}$ is the set of ID classes, and $n$ is the number of instances in $\mathcal{D}_{in}^{train}$. Let $\mathcal{D}^{test}$ denote the test set, which consists of ID test set $\mathcal{D}_{in}^{test}$ from ID distribution $P(\mathcal{X})$ and OOD test set $\mathcal{D}_{out}^{test}$ from OOD distribution $Q(\mathcal{X})$, $P(\mathcal{X}) \neq Q(\mathcal{X})$. The goal of OOD detection is to define a decision function $G$ such that for a given test input $\boldsymbol{x} \in \mathcal{D}^{test}$,

$$G(\mathrm{x}) = \begin{cases} 0 & \text{if } \boldsymbol{x} \sim Q(\mathcal{X}), \\ 1 & \text{if } \boldsymbol{x} \sim P(\mathcal{X}), \end{cases} \tag{1}$$

where $G(\boldsymbol{x}) = 1$ means that $\boldsymbol{x}$ is ID data and $G(\boldsymbol{x}) = 0$ means that $\boldsymbol{x}$ is OOD data.

## 2.2 MASTERED CRITERION

Prior work demonstrates that individual instances provide varying degrees of information and exert inconsistent influence on model learning throughout training (Hong et al., 2024). Consequently, instances that are well-learned early in training may contribute less to further performance improvement and can even induce overfitting if repeatedly used, ultimately degrading OOD detection. To mitigate overfitting on such mastered instances, it is essential to employ a simple and computationally efficient criterion to assess the model's learning status for each instance.

To efficiently identify which and when an instance is mastered, we construct a criterion based on the sample's uncertainty itself and the second-order difference of sample's uncertainty, which only relies on forward propagation. Intuitively, the uncertainty of a mastered instance should be relatively low and stable. When an instance $i$ is well fitted by the current model parameters $w^{(t)}$ or is insensitive to their recent update, the associated uncertainty $U_i(w^{(t)})$ will be small and reach a plateau. The sample's uncertainty itself and the second-order difference of the uncertainty will approach zero. To formalize this, when the sample's uncertainty itself and the second-order difference of the uncertainty for sample $i$ falls beneath a specified small positive threshold $\delta_1$ and $\delta_2$, sample $i$ is considered to be mastered by the model parameters $w$ and state $t$, which can be expressed as:

$$U_i(w^{(t)}) < \delta_1 \ \ \& \ \ \Delta^2 U_i(w^{(t)}) < \delta_2 \,. \tag{2}$$

To demonstrate the effectiveness of the mastered criteria, we experimentally tracked the number of instances that meet the mastered criteria during the training process. The mastered criteria enable adaptive sample selection throughout the learning process, allowing the model to dynamically adjust the size of the training set participating in backpropagation according to the evolving requirements. During the initial stages of training, the model has scarcely learned any instances, so the mastered criteria retain most instances for backpropagation, as almost every sample can provide useful information. As training progresses, the model gradually masters more instances, leading to an adaptive decrease in the number of retained training instances commensurate with the training progress. The mastered criteria adaptively remove these fully learned redundant instances from backpropagation, enabling the model to focus on the remaining samples and mitigating overfitting on the mastered samples.

## 2.3 UNCERTAINTY-DRIVEN RECTIFIED BACKPROPAGATION

Building upon the mastered criterion, we introduce uncertainty-driven rectified backpropagation strategy, allowing the model to stop backpropagation on an instance once it has been mastered. The core of out-of-distribution (OOD) detection lies in quantifying sample uncertainty. During the training phase, the uncertainty of training samples progressively decreases as the model optimizes. Consequently, uncertainty itself can serve as a metric to evaluate the model's mastery level of samples. According to (Yuan et al., 2025), different instances may have different optimal status due to factors such as sample complexity (Wang et al., 2020) and noise (Zhang et al., 2021). This poses a challenge in simply using uncertainty to construct mastered criterion for rectified backpropagation.

Therefore, we use the second-order difference of uncertainty to help identify the mastered instances, which quantifies the rate of change in the uncertainty for sample $i$ across three consecutive epochs, $t^{th}$, $(t-1)^{th}$, and $(t-2)^{th}$ training epochs. The second-order difference of uncertainty is defined as:

$$\Delta^2 U_i(w^{(t)}) = [U_i(w^{(t)}) - U_i(w^{(t-1)})] - [U_i(w^{(t-1)}) - U_i(w^{(t-2)})]. \tag{3}$$

By quantifying the rate of change in the uncertainty for each instance around the current parameters $w^{(t)}$, the second-order difference effectively captures the stability of the uncertainty function.

Accordingly, an instance $i$ is considered mastered when the cumulative magnitude of uncertainty itself and second-order differences of uncertainty falls beneath specified small positive thresholds $\delta_1$ and $\delta_2$, which is formally expressed as:

$$U_i(w^{(t)}) < \delta_1 \quad \& \quad |\Delta^2 U_i(w^{(t)})| < \delta_2. \tag{4}$$

Ultimately, uncertainty-driven rectified backpropagation consists of two key stages: filtering out mastered instances in the full training-set through forward propagation, and removing these mastered instances from backpropagation. Since different classes exhibit varying optimization progress, using a fixed threshold to determine when to exclude an instance from backpropagation is theoretically unsound. Accordingly, we develop an adaptive threshold by incorporating class-informed statistics to determine when to exclude an instance.

## 2.4 Uncertainty-driven Adaptively Rectified Backpropagation

Building on (Guo & Li, 2022), prior work on class-imbalanced semi-supervised learning shows that different classes exhibit varying learning progress; this motivated an adaptive-threshold mechanism for reliable sample selection to address imbalance. This raises the question: in class-balanced settings, does differential learning progress across classes still emerge? We find that it does. Specifically, we statistically analyzed the variance of uncertainty values across classes during training. Our analysis reveals that even in class-balanced scenarios, classes differ in learning difficulty: easy classes tend to have low uncertainty variance, while hard classes exhibit substantially higher variance. As reported in (Wei et al., 2024), models struggle to distinguish OOD data from samples belonging to high-variance classes. This observation motivates our key insight: impose a variance-consistency constraint during optimization to limit excessive inter-class variance divergence.

To expedite validation, we integrate the variance consistency constraint into CIDER (Ming et al., 2022b). Since it builds upon supervised contrastive learning, it provides a natural framework for incorporating variance consistency constraints. The implementation involves three key steps: (1) Computing Euclidean distances between all samples and their class prototypes within each batch to estimate per-class variance; (2) Constructing a variance set by collecting these class-wise variance measurements; (3) Formulating and minimizing a

Table 1: Comparison of CIDER (with variance consistency constraint, called VC) on CIFAR-10. All values are percentages, and OOD detection results are averaged over multiple OOD test datasets. $\uparrow$ indicates that larger values are better, while $\downarrow$ indicates that smaller values are better.

| Method | CIFAR-10 | | | |
| | Near-OOD | | Far-OOD | |
| | FPR95$\downarrow$ | AUROC$\uparrow$ | FPR95$\downarrow$ | AUROC$\uparrow$ |
| CIDER | 30.89 | 90.86 | 20.79 | 94.78 |
| CIDER + VC | 30.37 | 91.35 | 18.31 | 95.66 |

global variance consistency constraint based on the variance set statistics to validate our hypothesis. The results can be found in Table 1, which demonstrates that incorporating the Variance Consistency (VC) Constraint enhances CIDER's OOD detection performance consistently across both near-OOD and far-OOD scenarios.

Since different classes exhibit varying optimization progress, using a fixed threshold to determine when to exclude an instance from backpropagation is theoretically unsound. Based on the above the analysis and insight, we propose to integrate the VC constraint into Rectified Backpropagation. However, the direct integration of the VC constraint into Rectified Backpropagation presents several challenges: Three key challenges emerge: (1) Fundamental conflict exists between VC constraint and Rectified Backpropagation, since VC requires full-batch samples while Rectified Backpropagation selectively removes samples during backpropagation; (2) Non-contrastive baselines incur high computational costs when directly computing VC constraints; (3) Batch-level VC estimation becomes unreliable in many-class benchmarks.

To address the above challenges, we develop an adaptive threshold by incorporating class-informed statistics to determine when to exclude an instance from backpropagation. The static thresholds $\delta_1$ and $\delta_2$ in Eq. 4 cannot account for class-specific learning characteristics. We propose class-adaptive thresholds by incorporating

- Per-class variance of uncertainty: $\mathrm{Var}[U_i^{y_i}(w^{(t)})]$;

- Per-class variance of second-order difference of the uncertainty: $\mathrm{Var}[\Delta^2 U_i^{y_i}(w^{(t)})]$.

For class $y$ at training step $t$, define adaptive thresholds:

$$
\begin{aligned}
\delta_1^y &= \delta_1 + \gamma \delta_1 \left( 1 - \bar{\mathrm{V}}\mathrm{ar}[U^y(w^{(t)})] \right), \\
\delta_2^y &= \delta_2 + \gamma \delta_2 \left( 1 - \bar{\mathrm{V}}\mathrm{ar}[\Delta^2 U^y(w^{(t)})] \right),
\end{aligned}
\tag{5}
$$

where $\gamma$ is a hypeparameter and

$$
\begin{aligned}
\bar{\mathrm{V}}\mathrm{ar}[U^y(w^{(t)})] &= \frac{\mathrm{Var}[U^y(w^{(t)})] - \min\limits_{j\in\mathcal{Y}} \mathrm{Var}[U^j(w^{(t)})]}{\max\limits_{j\in\mathcal{Y}} \mathrm{Var}[U^j(w^{(t)})] - \min\limits_{j\in\mathcal{Y}} \mathrm{Var}[U^j(w^{(t)})]}, \\
\bar{\mathrm{V}}\mathrm{ar}[\Delta^2 U^y(w^{(t)})] &= \frac{\mathrm{Var}[\Delta^2 U^y(w^{(t)})] - \min\limits_{j\in\mathcal{Y}} \mathrm{Var}[\Delta^2 U^j(w^{(t)})]}{\max\limits_{j\in\mathcal{Y}} \mathrm{Var}[\Delta^2 U^j(w^{(t)})] - \min\limits_{j\in\mathcal{Y}} \mathrm{Var}[\Delta^2 U^j(w^{(t)})]}.
\end{aligned}
\tag{6}
$$

We then redefine the mastered criterion in Eq. 4 as follows:

$$
U_i(w^{(t)}) < \delta_1^{y_i} \quad \& \quad |\Delta^2 U_i(w^{(t)})| < \delta_2^{y_i}.
\tag{7}
$$

where $y_i$ denotes the class label of sample $i$. When the sample's uncertainty itself and the second-order difference of the uncertainty for sample $i$ falls beneath a specified small positive threshold $\delta_1^{y_i}$ and $\delta_2^{y_i}$, sample $i$ is considered to be mastered by the model parameters $w$ and state $t$. The detailed process of our proposed method can be seen in Algorithm 1.

---

**Algorithm 1** Uncertainty-driven Adaptively Rectified Backpropagation (UARB)

---

**Require:** Full training-set $\mathcal{D}_{in}^{train}$, model $f_\theta$, threshold $\delta_1, \delta_2$, max epochs $T$
1: Initialize model parameters $w^{(0)}$
2: **for** $t = 1$ to $T$ **do**
3:     Forward pass on model $f$ to compute uncertainty $U_i(w^{(t)})$ for each sample $i \in \mathcal{D}_{in}^{train}$
4:     Calculate second-order differences of uncertainty: $\Delta^2 U_i(w^{(t)})$ based on Eq. 3
5:     Calculate the adaptive thresholds $\delta_1^y, \delta_2^y$ for class $y$ based on Eq. 5
6:     Identify *mastered* instances by the criterion in Eq. 7
7:     Stop backpropagation of mastered instances by setting the loss of mastered instances to zero
8:     Update model parameters $w^{(t)}$ by backpropagation
9: **end for**

---

## 3 EXPERIMENTS

In this section, we evaluate UARB on various OOD detection tasks. We first verify the effectiveness of UARB on CIFAR benchmarks and ImageNet benchmarks. Then, we provide further analysis.

### 3.1 EVALUATION ON CIFAR BENCHMARKS

**ID Datasets:** We use CIFAR-10 and CIFAR-100 (Krizhevsky et al., 2009) as ID data.

**OOD Datasets:** For the OOD datasets, we use six common benchmarks as used in previous study under two OOD detection scenarios, including near-OOD detection and far OOD detection. The

Table 2: Comparison of OOD detection methods (with our proposed UARB) on CIFAR benchmarks. All values are percentages, and OOD detection results are averaged over multiple OOD test datasets. Detailed results for each OOD dataset are provided in Appendix. ↑ indicates that larger values are better, while ↓ indicates that smaller values are better.

| Method | CIFAR-10 | | | | CIFAR-100 | | | |
| | Near-OOD | | Far-OOD | | Near-OOD | | Far-OOD | |
| | FPR95↓ | AUROC↑ | FPR95↓ | AUROC↑ | FPR95↓ | AUROC↑ | FPR95↓ | AUROC↑ |
|---|---|---|---|---|---|---|---|---|
| MSP | 47.41 ± 0.99 | 88.18 ± 0.34 | 29.54 ± 1.91 | 91.01 ± 0.47 | 55.15 ± 0.13 | 80.27 ± 0.11 | 58.57 ± 1.50 | 78.06 ± 0.88 |
| MSP + UM | 39.06 ± 1.76 | 87.98 ± 0.45 | 26.08 ± 1.20 | 91.79 ± 0.48 | 55.27 ± 0.24 | 80.18 ± 0.16 | 57.47 ± 1.14 | 78.76 ± 0.50 |
| MSP + ours | 32.47 ± 1.00 | 89.41 ± 0.25 | 21.28 ± 0.75 | 92.98 ± 0.36 | 55.00 ± 0.34 | 80.45 ± 0.09 | 55.30 ± 2.17 | 79.25 ± 0.71 |
| Energy | 61.89 ± 1.48 | 87.60 ± 0.47 | 40.32 ± 3.99 | 91.40 ± 0.75 | 55.99 ± 0.27 | 80.88 ± 0.08 | 56.02 ± 1.58 | 80.08 ± 0.99 |
| Energy + UM | 45.84 ± 3.17 | 88.59 ± 0.33 | 25.48 ± 2.77 | 93.60 ± 0.36 | 57.36 ± 0.48 | 80.35 ± 0.21 | 55.64 ± 1.09 | 80.28 ± 0.76 |
| Energy + ours | 34.07 ± 0.61 | 90.93 ± 0.13 | 18.28 ± 0.54 | 95.31 ± 0.23 | 55.54 ± 0.54 | 81.06 ± 0.17 | 53.05 ± 2.70 | 81.06 ± 1.08 |
| KNN | 33.56 ± 0.73 | 90.77 ± 0.28 | 22.61 ± 1.21 | 93.31 ± 0.29 | 61.59 ± 0.83 | 80.26 ± 0.10 | 55.02 ± 3.43 | 82.38 ± 1.10 |
| KNN + UM | 41.88 ± 2.79 | 87.48 ± 0.82 | 27.23 ± 0.48 | 91.50 ± 0.30 | 64.16 ± 0.46 | 79.81 ± 0.23 | 51.54 ± 2.08 | 83.36 ± 0.84 |
| KNN + ours | 34.14 ± 0.59 | 90.20 ± 0.14 | 21.34 ± 0.82 | 93.85 ± 0.63 | 60.91 ± 1.18 | 80.41 ± 0.13 | 51.55 ± 2.94 | 82.91 ± 1.01 |
| ReAct | 59.78 ± 3.10 | 87.90 ± 0.42 | 39.25 ± 4.36 | 91.40 ± 0.68 | 56.59 ± 0.31 | 80.80 ± 0.11 | 53.78 ± 1.18 | 80.65 ± 0.72 |
| ReAct + UM | 47.52 ± 6.73 | 88.37 ± 0.69 | 26.99 ± 4.61 | 93.36 ± 0.70 | 58.32 ± 0.21 | 80.15 ± 0.17 | 54.10 ± 0.57 | 80.77 ± 0.55 |
| ReAct + ours | 34.31 ± 0.86 | 90.84 ± 0.09 | 18.81 ± 1.13 | 95.22 ± 0.34 | 56.39 ± 0.54 | 80.93 ± 0.17 | 50.34 ± 2.38 | 81.65 ± 1.04 |
| Relation | 36.62 ± 0.92 | 89.97 ± 0.31 | 25.52 ± 1.36 | 92.56 ± 0.40 | 60.80 ± 0.99 | 80.71 ± 0.13 | 56.55 ± 3.25 | 80.94 ± 0.96 |
| Relation + UM | 43.54 ± 5.38 | 87.56 ± 0.79 | 26.82 ± 1.24 | 91.88 ± 0.19 | 61.14 ± 0.30 | 80.54 ± 0.05 | 52.66 ± 1.63 | 81.79 ± 0.50 |
| Relation + ours | 38.28 ± 0.99 | 88.98 ± 0.59 | 23.90 ± 0.97 | 92.99 ± 0.08 | 60.89 ± 0.94 | 80.83 ± 0.06 | 52.77 ± 3.20 | 81.79 ± 0.98 |
| Fdbd | 34.22 ± 0.79 | 90.68 ± 0.26 | 22.40 ± 1.30 | 93.43 ± 0.38 | 56.61 ± 0.73 | 81.07 ± 0.09 | 55.18 ± 2.12 | 80.23 ± 0.91 |
| Fdbd + UM | 37.70 ± 2.06 | 89.38 ± 0.51 | 24.30 ± 0.93 | 93.21 ± 0.22 | 57.72 ± 0.55 | 80.80 ± 0.11 | 51.60 ± 1.04 | 81.25 ± 0.40 |
| Fdbd + ours | 32.63 ± 0.60 | 90.74 ± 0.19 | 20.09 ± 0.84 | 94.32 ± 0.52 | 56.39 ± 0.81 | 81.25 ± 0.10 | 51.31 ± 2.83 | 81.26 ± 1.06 |
| Nci | 47.07 ± 0.99 | 88.75 ± 0.26 | 29.35 ± 1.56 | 91.55 ± 0.40 | 55.76 ± 0.60 | 81.00 ± 0.05 | 51.44 ± 1.52 | 81.96 ± 0.63 |
| Nci + UM | 46.71 ± 3.31 | 87.24 ± 0.76 | 29.96 ± 1.09 | 91.68 ± 0.14 | 56.59 ± 0.18 | 80.87 ± 0.18 | 48.44 ± 0.97 | 82.73 ± 0.55 |
| Nci + ours | 39.26 ± 1.40 | 88.98 ± 0.33 | 24.01 ± 0.92 | 93.22 ± 0.42 | 56.74 ± 0.28 | 81.10 ± 0.02 | 48.43 ± 1.96 | 82.50 ± 0.63 |

near-OOD group contains CIFAR-100/10 and TinyImageNet (Le & Yang, 2015), while the far-OOD group consists of MNIST, SVHN (Netzer et al., 2011), Texture (Cimpoi et al., 2014), and Places365 (Zhou et al., 2017). For the realistic outlier dataset, we follow the OpenOOD standards and use TinyImageNet-597 (Le & Yang, 2015), which has no category overlap with CIFAR-10/100 and the OOD test sets.

**Evaluation Metrics:** We report two widely adopted metrics: (1) FPR-95%-TPR (FPR95), which can be interpreted as the probability that a negative (OOD data) sample is classified as an ID prediction when the TPR is as high as 95%; (2) AUROC, which measures the area under the receiver operating characteristic curve. The ROC curve depicts the relationship between TPR and false positive rate.

**Training Details:** In line with OpenOOD (Zhang et al., 2023a), we adopt ResNet-18 (He et al., 2016) as the backbone architecture. Models are trained with stochastic gradient descent (SGD) for 100 epochs, using a learning rate of 0.1 with cosine annealing decay schedule (Loshchilov & Hutter, 2017), momentum of 0.9, and weight decay of $5 \times 10^{-4}$. $\delta_1$ and $\delta_2$ are 0.01 and 0.001. We run each trial three times. The batch size is set to 128 for CIFAR-10/100.

**Baselines:** We compare our proposed method with a suite of competitive baselines, including: **MSP** (Hendrycks & Gimpel, 2016b), **Energy** (Liu et al., 2020), **KNN** (Sun et al., 2022), **ReAct** (Sun et al., 2021), **Relation** (Kim et al., 2023), **Fbdb** (Liu & Qin, 2023), **Nci** (Liu & Qin, 2025). Moreover, we compare the most relevant method **UM** (Zhu et al., 2023).

**UARB Boosts Far OOD Detection:** The detailed experiments for comparison with mostly recent methods are conducted, and the results are listed in Table 2. UARB can be applied to existing methods like MSP and boost their performance significantly. For CIFAR-10 as the ID dataset, UARB improves MSP's far-OOD detection performance (AUROC: from 91.01% to 92.98%). Similarly for CIFAR-100, UARB reduces FPR95 on far-OOD detection from 58.57% to 55.30%. These results show the effectiveness of UARB for far-OOD detection.

**UARB Boosts Near OOD Detection:** While near-OOD detection proves more challenging than far-OOD tasks, UARB still enhances baseline performance as shown in Table 2. For CIFAR-10 as ID data, UARB improves MSP's near-OOD AUROC from 88.18% to 89.41%, and reduces MSP's near-OOD FPR95 from 47.41% to 32.47%. These results demonstrate UARB's effectiveness on near-OOD detection, though the smaller improvements compared to far-OOD (AUROC improved by 1.23% vs 1.97%) reflect the intrinsic difficulty of near-OOD scenarios.

Table 3: Comparison (FPR95↓) on ImageNet benchmark. All values are percentages.

| Method | Ssb_hard | Ninco | Near-Avg | iNaturalist | Textures | Openimage-o | Far-Avg |
|---|---|---|---|---|---|---|---|
| MSP | 35.26 ± 0.40 | 31.72 ± 0.61 | 33.49 ± 0.50 | 21.39 ± 0.08 | 23.70 ± 0.46 | 25.02 ± 0.31 | 23.37 ± 0.21 |
| MSP+UM | 36.04 ± 0.35 | 32.19 ± 0.20 | 34.12 ± 0.10 | 21.64 ± 0.57 | 23.85 ± 0.83 | 26.10 ± 1.03 | 23.87 ± 0.81 |
| MSP+ours | 34.56 ± 0.73 | 30.49 ± 0.22 | 32.53 ± 0.45 | 20.59 ± 0.92 | 23.23 ± 0.79 | 23.82 ± 0.65 | 22.55 ± 0.78 |
| Energy | 39.01 ± 0.24 | 34.52 ± 1.23 | 36.76 ± 0.73 | 18.58 ± 0.35 | 22.07 ± 1.32 | 22.56 ± 0.58 | 21.07 ± 0.72 |
| Energy+UM | 42.77 ± 0.75 | 38.77 ± 0.70 | 40.77 ± 0.71 | 19.91 ± 1.20 | 23.17 ± 2.04 | 25.69 ± 1.70 | 22.92 ± 1.36 |
| Energy+ours | 38.59 ± 1.09 | 33.32 ± 0.90 | 35.96 ± 0.50 | 17.35 ± 0.82 | 20.18 ± 0.53 | 22.84 ± 0.84 | 20.12 ± 0.66 |
| KNN | 34.03 ± 0.58 | 26.50 ± 0.58 | 30.27 ± 0.58 | 15.85 ± 0.42 | 10.10 ± 0.27 | 16.99 ± 0.53 | 14.32 ± 0.40 |
| KNN+UM | 34.83 ± 0.82 | 27.84 ± 0.81 | 31.33 ± 0.46 | 17.11 ± 1.26 | 10.07 ± 0.40 | 18.06 ± 0.61 | 15.08 ± 0.48 |
| KNN+ours | 32.98 ± 0.58 | 26.80 ± 0.23 | 29.89 ± 0.33 | 14.87 ± 0.56 | 9.72 ± 0.18 | 16.74 ± 0.34 | 13.78 ± 0.17 |
| ReAct | 38.89 ± 0.66 | 32.98 ± 0.58 | 35.93 ± 0.37 | 17.70 ± 0.34 | 20.41 ± 1.19 | 21.56 ± 0.57 | 19.89 ± 0.70 |
| ReAct+UM | 43.01 ± 1.75 | 36.55 ± 0.49 | 39.78 ± 0.98 | 18.80 ± 1.07 | 19.96 ± 1.23 | 23.56 ± 1.49 | 20.77 ± 1.07 |
| ReAct+ours | 38.84 ± 1.05 | 32.16 ± 0.83 | 35.50 ± 0.49 | 17.02 ± 1.11 | 18.80 ± 0.85 | 21.99 ± 1.39 | 19.27 ± 0.93 |
| Relation | 33.66 ± 0.27 | 27.16 ± 0.53 | 30.41 ± 0.39 | 16.09 ± 0.55 | 13.07 ± 0.04 | 18.61 ± 0.59 | 15.92 ± 0.34 |
| Relation+UM | 33.96 ± 0.38 | 27.69 ± 0.16 | 30.82 ± 0.11 | 15.30 ± 0.39 | 12.83 ± 0.23 | 18.61 ± 0.61 | 15.58 ± 0.37 |
| Relation+ours | 32.81 ± 0.24 | 26.58 ± 0.38 | 29.70 ± 0.30 | 14.83 ± 0.71 | 12.71 ± 0.04 | 18.07 ± 0.36 | 15.20 ± 0.35 |
| Fdbd | 32.83 ± 0.29 | 25.41 ± 0.44 | 29.12 ± 0.33 | 13.96 ± 0.45 | 12.31 ± 0.38 | 17.17 ± 0.70 | 14.48 ± 0.38 |
| Fdbd+UM | 33.24 ± 0.23 | 25.26 ± 0.35 | 29.25 ± 0.10 | 13.59 ± 0.37 | 12.50 ± 0.09 | 17.19 ± 0.18 | 14.42 ± 0.19 |
| Fdbd+ours | 32.47 ± 0.74 | 24.46 ± 0.29 | 28.47 ± 0.51 | 13.13 ± 0.33 | 11.93 ± 0.44 | 16.40 ± 0.40 | 13.82 ± 0.34 |
| Nci | 33.27 ± 0.33 | 27.32 ± 0.69 | 30.30 ± 0.46 | 15.63 ± 0.23 | 12.65 ± 0.44 | 18.19 ± 0.51 | 15.49 ± 0.28 |
| Nci+UM | 34.31 ± 0.28 | 27.66 ± 0.30 | 30.99 ± 0.03 | 15.17 ± 0.48 | 12.62 ± 0.20 | 18.19 ± 0.15 | 15.33 ± 0.15 |
| Nci+ours | 32.77 ± 0.34 | 26.40 ± 0.49 | 29.59 ± 0.34 | 14.80 ± 0.59 | 12.84 ± 0.20 | 17.85 ± 0.59 | 15.17 ± 0.43 |

**UARB Is Compatible with Various OOD Scoring Functions:** Since UARB is a simple module readily pluggable into the existing methods, we add it to various OOD scoring methods. We report the results of adding UARB to MSP, Energy, KNN, ReAct, Relation, Fbdb, and Nci, and it shows consistent improvement in Table 2. For CIFAR-10 as ID data in far-OOD detection, UARB achieves significant FPR95 reductions across methods: MSP (-4.41%), Energy (-6.80%), KNN (-4.77%), ReAct (-4.40%), Relation (-9.21%), Fdbd (-5.45%), and Nci (-10.35%). UARB can still achieve a consistent improvement on various OOD scoring functions.

**Comparison with the Most Relevant UM Method:** UM is the method most relevant to our method, which also mitigates overfitting from a data perspective. UM determines whether a sample is over-fitted based on its loss, and then "forgets" such samples by increasing their loss. In contrast, our approach leverages uncertainty to evaluate whether a sample has been fully mastered for OOD detection. We then utilize zero-order and second-order statistics of uncertainty to determine if a sample has been thoroughly learned. For samples that are already mastered, we rectify backpropagation during training. Furthermore, we account for variations in learning progress across different classes by adaptively rectified backpropagation strategy tailored to inter-class differences. According to Table 2, it can be observed that our method outperforms UM in almost all aspects. These results demonstrate the effectiveness of our approach.

## 3.2 Evaluation on ImageNet Benchmarks

**ID Datasets:** We use ImageNet (Deng et al., 2009)) as the ID dataset.

**OOD Datasets:** For the OOD datasets, we use five common benchmarks as used in previous study under two OOD detection scenarios, including near-OOD detection and far OOD detection. The near-OOD group contains SSB-hard (Vaze et al., 2021) and NINCO (Bitterwolf et al., 2023), while the far-OOD group consists of iNaturalist (Van Horn et al., 2018), Texture (Cimpoi et al., 2014), and OpenImage-O (Wang et al., 2022).

**Training Details:** In line with OpenOOD (Zhang et al., 2023a), we adopt ResNet-18 (He et al., 2016) as the backbone. Models are trained with stochastic gradient descent (SGD) for 100 epochs, using a learning rate of 0.1 with cosine annealing decay schedule, momentum of 0.9, and weight decay of $5 \times 10^{-4}$. We run each trial three times. The batch size is set to 256 for ImageNet.

**Results:** The results for ImageNet are listed in Table 3. It reveals that the model with UARB achieves great improvement in near-OOD detection and far-OOD detection. UARB is orthogonal to OOD detection baselines. When integrated with UARB, most of these methods exhibit improved OOD detection performance. This demonstrates the efficacy of the rectified backpropagation mech-

Table 4: Comparison of OOD detection methods (with our proposed UARB, URB) on CIFAR-10 benchmark. All values are percentages.

| Method | MNIST | | SVHN | | Texture | | Places365 | |
|---|---|---|---|---|---|---|---|---|
| | FPR95↓ | AUROC↑ | FPR95↓ | AUROC↑ | FPR95↓ | AUROC↑ | FPR95↓ | AUROC↑ |
| Energy | 28.40 ± 3.34 | 93.51 ± 0.59 | 36.60 ± 11.38 | 91.54 ± 1.91 | 43.33 ± 1.81 | 90.78 ± 0.34 | 52.94 ± 1.80 | 89.75 ± 0.41 |
| Energy + URB | 14.55 ± 3.33 | 96.45 ± 0.96 | 11.90 ± 2.12 | 97.45 ± 0.33 | 23.56 ± 2.48 | 93.90 ± 0.99 | 26.51 ± 0.82 | 93.38 ± 0.43 |
| Energy + UARB | 11.54 ± 1.38 | 97.22 ± 0.45 | 11.47 ± 2.53 | 97.17 ± 1.02 | 23.12 ± 0.58 | 93.75 ± 0.21 | 26.98 ± 1.47 | 93.11 ± 0.52 |
| ReAct | 36.84 ± 7.29 | 92.30 ± 1.02 | 36.47 ± 9.09 | 91.42 ± 1.45 | 37.99 ± 5.40 | 91.42 ± 0.64 | 45.70 ± 2.64 | 90.45 ± 0.45 |
| ReAct + URB | 17.69 ± 3.36 | 95.66 ± 0.98 | 13.93 ± 0.42 | 96.92 ± 0.29 | 26.60 ± 5.82 | 93.31 ± 1.56 | 26.75 ± 1.55 | 93.35 ± 0.55 |
| ReAct + UARB | 13.15 ± 2.46 | 96.92 ± 0.66 | 12.67 ± 3.22 | 96.90 ± 1.25 | 23.17 ± 0.46 | 93.86 ± 0.07 | 26.24 ± 2.47 | 93.22 ± 0.84 |

Table 5: FPR95($\downarrow$) comparison on CIFAR-10 benchmark. All values are percentages.

| | $U$ | $\Delta^2 U$ | CIFAR-100 | TIN | Near-Avg | MNIST | SVHN | Texture | Places365 | Far-Avg |
|---|---|---|---|---|---|---|---|---|---|---|
| Energy | - | - | 67.57 ± 0.98 | 56.21 ± 2.24 | 61.89 ± 1.48 | 28.40 ± 3.34 | 36.60 ± 11.38 | 43.33 ± 1.81 | 52.94 ± 1.80 | 40.32 ± 3.99 |
| | ✓ | - | 40.85 ± 1.60 | 30.69 ± 2.36 | 35.77 ± 1.87 | 12.36 ± 2.06 | 12.97 ± 1.39 | 22.57 ± 2.37 | 28.04 ± 2.24 | 18.98 ± 0.22 |
| | - | ✓ | 38.43 ± 0.58 | 30.07 ± 0.46 | 34.25 ± 0.38 | 13.17 ± 2.12 | 13.66 ± 3.84 | 23.53 ± 1.07 | 28.27 ± 1.74 | 19.66 ± 0.74 |
| | ✓ | ✓ | 38.53 ± 0.69 | 29.61 ± 0.82 | 34.07 ± 0.61 | 11.54 ± 1.38 | 11.47 ± 2.53 | 23.12 ± 0.58 | 26.98 ± 1.47 | 18.28 ± 0.54 |
| ReAct | - | - | 64.95 ± 3.63 | 54.60 ± 2.56 | 59.78 ± 3.10 | 36.84 ± 7.29 | 36.47 ± 9.09 | 37.99 ± 5.40 | 45.70 ± 2.64 | 39.25 ± 4.36 |
| | ✓ | - | 43.76 ± 4.15 | 34.23 ± 4.04 | 39.00 ± 4.08 | 13.78 ± 1.10 | 14.75 ± 1.98 | 24.88 ± 4.31 | 26.65 ± 1.47 | 20.01 ± 1.05 |
| | - | ✓ | 47.49 ± 4.99 | 38.62 ± 3.33 | 43.05 ± 4.02 | 18.59 ± 4.94 | 17.99 ± 3.50 | 27.35 ± 2.12 | 27.76 ± 2.91 | 22.92 ± 0.49 |
| | ✓ | ✓ | 38.75 ± 1.37 | 29.88 ± 0.72 | 34.31 ± 0.86 | 13.15 ± 2.46 | 12.67 ± 3.22 | 23.17 ± 0.46 | 26.24 ± 2.47 | 18.81 ± 1.13 |

anisms, which excludes the mastered instance to effectively push the model beyond its intrinsic OOD detection capability limits.

### 3.3 FURTHER ANALYSIS

**Effectiveness Analysis of Adaptive Strategies.** We develop an adaptive threshold by incorporating class-informed statistics to determine when to exclude an instance from backpropagation. To validate the effectiveness of this adaptive strategy, we compared the performance of the proposed method with and without the adaptive strategy. The experimental results are shown in Table 4. We conducted validation on the in-distribution (ID) dataset CIFAR-10, reporting the results of UARB with the adaptive strategy and RB without it. The methods were deployed on both Energy and ReAct scoring function. Experimental results demonstrate that employing the adaptive strategy leads to performance improvements across multiple OOD datasets. These results confirm the effectiveness of the adaptive strategy.

**Effectiveness Analysis of Mastered Criterion.** According to Eq. 7, our method considers an instance as mastered if the zero-order and second-order differences of its uncertainty value remain within a small range around zero, which offer a more effective measure of an instance's learning status. To validate the effectiveness of zero-order $U$ and second-order $\Delta^2 U$ statistics, we conducted ablation studies by removing each of them respectively. The experimental results are presented in Table 5. We report the FPR95 under both near-OOD and far-OOD scenarios, along with their corresponding average results (near-avg and far-avg). Experimental results demonstrate that utilizing both statistics for decision-making achieves better performance, validating the effectiveness of zero-order and second-order metrics.

**Analysis of Hypeparameter Sensitivity.** $\gamma$ is the hyperparameter in Eq. 5 used to balance the class-aware adaptive term. $\gamma = 0$ indicates that the adaptive strategy is not applied. We evaluated the results on CIFAR-10 about average FPR95 over Far-OOD scenario with $\gamma$ set to 0.02, 0.05, and 0.1, and the experimental results are shown in Table 6. The experimental results

Table 6: Sensitivity analysis (FPR95$\downarrow$) of $\gamma$.

| Method | $\gamma = 0$ | $\gamma = 0.02$ | $\gamma = 0.05$ | $\gamma = 0.1$ |
|---|---|---|---|---|
| MSP +UARB | 56.11 ± 2.53 | 55.30 ± 2.17 | 55.48 ± 0.46 | 56.70 ± 0.47 |
| Energy +UARB | 53.92 ± 2.31 | 53.05 ± 2.70 | 53.88 ± 0.53 | 54.91 ± 1.05 |
| KNN +UARB | 51.82 ± 1.69 | 51.55 ± 2.94 | 51.14 ± 1.11 | 53.85 ± 0.46 |
| ReAct+UARB | 52.25 ± 1.82 | 50.34 ± 2.38 | 51.97 ± 0.74 | 53.11 ± 1.57 |
| Relation+UARB | 54.02 ± 2.34 | 52.77 ± 3.20 | 52.68 ± 1.06 | 55.45 ± 0.11 |
| Fdbd+UARB | 52.46 ± 2.76 | 51.31 ± 2.83 | 50.99 ± 1.58 | 53.78 ± 1.01 |
| Nci+UARB | 49.00 ± 1.57 | 48.43 ± 1.96 | 48.03 ± 1.60 | 49.69 ± 0.82 |

indicate that as $\gamma$ increases, our method demonstrates a consistent trend of initial decrease followed by an increase in FPR95 across all evaluation metrics. Compared to $\gamma = 0$, the performance of our method improves significantly at $\gamma = 0.02$ and $\gamma = 0.05$, confirming the effectiveness of the adaptive strategy. However, further increasing $\gamma$ leads to performance degradation. This suggests that excessive integration of the class-aware adaptive term interferes with the selection of mastered instances.

Table 7: FPR95 and AUROC results across multiple OOD datasets on large-scale ImageNet benchmark. EMA deontes Exponential Moving Average.

| Method | iNaturalist | | Texture | | Openimage-o | |
|---|---|---|---|---|---|---|
| | FPR95↓ | AUROC↑ | FPR95↓ | AUROC↑ | FPR95↓ | AUROC↑ |
| Energy | 18.58 ± 0.35 | 95.39 ± 0.25 | 22.07 ± 1.32 | 94.91 ± 0.12 | 22.56 ± 0.58 | 94.29 ± 0.05 |
| Energy + UARB | 17.35 ± 0.82 | 95.47 ± 0.15 | 20.18 ± 0.53 | 95.16 ± 0.05 | 22.84 ± 0.84 | 94.29 ± 0.09 |
| Energy + UARB + EMA | 17.16 ± 1.13 | 95.68 ± 0.39 | 19.85 ± 0.50 | 95.61 ± 0.09 | 21.39 ± 0.92 | 94.60 ± 0.18 |
| ReAct | 17.70 ± 0.34 | 95.52 ± 0.26 | 20.41 ± 1.19 | 95.34 ± 0.09 | 21.56 ± 0.57 | 94.97 ± 0.05 |
| ReAct + UARB | 17.02 ± 1.11 | 95.55 ± 0.21 | 18.80 ± 0.85 | 95.48 ± 0.12 | 21.99 ± 1.39 | 94.40 ± 0.16 |
| ReAct + UARB + EMA | 16.99 ± 1.55 | 95.72 ± 0.41 | 17.91 ± 0.71 | 95.91 ± 0.21 | 21.17 ± 0.99 | 94.70 ± 0.15 |

Table 8: Comparison (FPR95↓) under different uncertainty selection about $U$ on CIFAR-10.

| $U$ | CIFAR-100 | TIN | Near-Avg | MNIST | SVHN | Texture | Places365 | Far-Avg |
|---|---|---|---|---|---|---|---|---|
| None | 67.57 ± 0.98 | 56.21 ± 2.24 | 61.89 ± 1.48 | 28.40 ± 3.34 | 36.60 ± 11.38 | 43.33 ± 1.81 | 52.94 ± 1.80 | 40.32 ± 3.99 |
| -MSP | 38.53 ± 0.69 | 29.61 ± 0.82 | 34.07 ± 0.61 | 11.54 ± 1.38 | 11.47 ± 2.53 | 23.12 ± 0.58 | 26.98 ± 1.47 | 18.28 ± 0.54 |
| Loss | 39.06 ± 0.36 | 29.47 ± 0.47 | 34.27 ± 0.34 | 17.72 ± 1.78 | 13.50 ± 1.66 | 21.68 ± 2.76 | 27.30 ± 0.73 | 20.05 ± 0.94 |
| Entropy | 40.92 ± 2.30 | 32.74 ± 2.98 | 36.83 ± 1.67 | 16.17 ± 0.78 | 17.87 ± 1.16 | 25.89 ± 2.01 | 26.44 ± 0.64 | 21.59 ± 0.89 |

**Analysis of Model EMA.** Furthermore, we integrated Exponential Moving Average (EMA) into our method to enhance OOD detection performance. The approach was validated on the large-scale dataset ImageNet, and we reported FPR95 and AUROC results across multiple OOD datasets. Experimental results are shown in Table 7. The results demonstrate that the performance of our method UARB, was further improved after incorporating EMA. This enhancement can be attributed to EMA's ability to aggregate knowledge learned over multiple training epochs, thereby strengthening the model's robustness.

**Analysis under Different Uncertainty Scores.** In Eq. 3, $U$ denotes the uncertainty score, based on which we utilize its zero-order and second-order statistics to determine whether a sample has been mastered. Here, we compare the FPR95↓ of our method under different uncertainty scores on CIFAR-10, with the experimental results presented in Table 8. We present results respectively for the following scenarios: not using any uncertainty score (i.e., without applying UARB), and using -MSP, Loss, or Entropy as the uncertainty score. The experimental results demonstrate the effectiveness of employing uncertainty scores to exclude mastered instances for mitigating overfitting. However, the performance varies slightly across different uncertainty scores. Compared to Loss and Entropy, -MSP proves to be the most effective.

## 4 CONCLUSION

In this paper, we have proposed Uncertainty-driven Adaptively Rectified Backpropagation (UARB), a novel data-centric approach that addresses the critical challenge of overfitting in out-of-distribution (OOD) detection. Departing from existing model-centric strategies such as post-training distillation or pruning, UARB introduces an end-to-end training framework grounded in the principle of ceasing updates for instances that the model has already mastered. Our method leverages the consistency of both zero-order and second-order uncertainty statistics to accurately identify such mastered instances, providing a more stable measure of learning progress than single-moment criteria. Furthermore, UARB incorporates a class-aware adaptive threshold to dynamically determine when to exclude instances from backpropagation, accounting for the varied learning paces across different classes.

Extensive experimental evaluations demonstrate that UARB effectively enhances OOD detection performance by systematically mitigating overfitting. The proposed method offers a principled and practical solution for improving model reliability on unseen distributions, highlighting the significant potential of data-centric optimization in building more robust AI systems. Future work may explore extending this adaptive principle to other safety-critical applications beyond OOD detection.

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

## A APPENDIX

### A.1 RELATED WORKS

#### A.1.1 OUT-OF-DISTRIBUTION DETECTION

**Test-time OOD detection methods.** Test-time OOD detection methods have the advantage of being easy to use without modifying the training procedure and objective (Yang et al., 2021). The test-time approaches do not require retraining the model, performs well, and is easy to implement in the real world. Test-time OOD detection methods can be categorized into confidence-based, feature-based, distance-based, gradient-based, pruning-based, and activation-based methods. Confidence-based methods use the confidence score of a pre-trained classifier to detect OOD data. The underlying assumption is that the ID data should receive a high confidence score, while the OOD data should receive a low confidence score. MSP (Hendrycks & Gimpel, 2016a) directly uses the maximum SoftMax score to determine whether the test sample is an ID or OOD. ODIN (Liang et al., 2017) improves the SoftMax score by perturbing the input and applying temperature scaling to the logits. Energy (Liu et al., 2020) demonstrates that the Energy score (i.e., logsumexp of logits) outperforms the SoftMax score in distinguishing between ID and OOD data. Feature-based methods include GRAM (Sastry & Oore, 2020) and SHE (Zhang et al., 2023b). GRAM computes the gram matrix within the hidden layers. SHE uses the energy function defined in modern Hopfield networks. Distance-based methods consider OOD data to be farther away from the training set than ID data. Mahalanobis (Lee et al., 2018) calculates the minimum Mahalanobis distance between the test data and the class centroids of the training set as an OOD score. Gradient-based approaches (Huang et al., 2021) uses gradient statistics to calculate OOD score. Pruning-based methods prunes the weights of model to address overconfident prediction of OOD data. DICE (Sun & Li, 2022) prunes the weights of the classification layer to address overconfidence in the model's prediction of OOD data. Activation-based methods attempt to maximize the gap between ID and OOD data by truncating abnormally low or high activations. ReAct (Sun et al., 2021) observes that OOD inputs trigger abnormally high activations. BATS (Zhu et al., 2022; He et al., 2024) exhibits efficacy by truncating both abnormally low and abnormally high activations of each channel.

**Training-time OOD detection methods.** Unlike testing-time methods, training-time methods aim to mitigate overconfident predictions for OOD data during the training period. According to whether

Table 9: Comparison (FPR95↓) on CIFAR-10 benchmark. All values are percentages.

| Method | CIFAR-100 | TIN | Near-Avg | MNIST | SVHM | Texture | Places365 | Far-Avg |
|---|---|---|---|---|---|---|---|---|
| MSP | 52.70 ± 1.47 | 42.13 ± 2.26 | 47.41 ± 0.99 | 25.33 ± 1.50 | 23.77 ± 4.86 | 28.23 ± 0.23 | 40.83 ± 1.74 | 29.54 ± 1.91 |
| MSP+UM | 43.63 ± 1.57 | 34.49 ± 2.22 | 39.06 ± 1.76 | 31.55 ± 10.36 | 13.43 ± 1.72 | 25.85 ± 1.77 | 33.50 ± 3.75 | 26.08 ± 1.20 |
| MSP+ours | 34.99 ± 0.97 | 29.96 ± 1.03 | 32.47 ± 1.00 | 18.56 ± 2.17 | 13.67 ± 0.86 | 23.42 ± 0.92 | 29.48 ± 1.05 | 21.28 ± 0.75 |
| Energy | 67.57 ± 0.98 | 56.21 ± 2.24 | 61.89 ± 1.48 | 28.40 ± 3.34 | 36.60 ± 11.38 | 43.33 ± 1.81 | 52.94 ± 1.80 | 40.32 ± 3.99 |
| Energy+UM | 53.51 ± 3.84 | 38.17 ± 3.21 | 45.84 ± 3.17 | 24.29 ± 7.85 | 17.50 ± 8.53 | 30.47 ± 6.06 | 29.67 ± 5.18 | 25.48 ± 2.77 |
| Energy+ours | 38.53 ± 0.69 | 29.61 ± 0.82 | 34.07 ± 0.61 | 11.54 ± 1.38 | 11.47 ± 2.53 | 23.12 ± 0.58 | 26.98 ± 1.47 | 18.28 ± 0.54 |
| KNN | 37.44 ± 0.75 | 29.68 ± 0.80 | 33.56 ± 0.73 | 20.18 ± 0.98 | 20.19 ± 2.72 | 21.34 ± 0.56 | 28.73 ± 1.00 | 22.61 ± 1.21 |
| KNN+UM | 45.11 ± 2.37 | 38.65 ± 3.23 | 41.88 ± 2.79 | 20.00 ± 0.68 | 24.09 ± 4.42 | 30.44 ± 2.52 | 34.39 ± 3.45 | 27.23 ± 0.48 |
| KNN+ours | 37.66 ± 0.58 | 30.61 ± 0.64 | 34.14 ± 0.59 | 15.89 ± 2.45 | 18.56 ± 2.97 | 21.82 ± 0.98 | 29.09 ± 2.12 | 21.34 ± 0.82 |
| ReAct | 64.95 ± 3.63 | 54.60 ± 2.56 | 59.78 ± 3.10 | 36.84 ± 7.29 | 36.47 ± 9.09 | 37.99 ± 5.40 | 45.70 ± 2.64 | 39.25 ± 4.36 |
| ReAct+UM | 54.29 ± 6.29 | 40.74 ± 7.38 | 47.52 ± 6.73 | 29.30 ± 6.62 | 20.88 ± 12.03 | 29.66 ± 9.02 | 28.12 ± 4.10 | 26.99 ± 4.61 |
| ReAct+ours | 38.75 ± 1.37 | 29.88 ± 0.72 | 34.31 ± 0.86 | 13.15 ± 2.46 | 12.67 ± 3.22 | 23.17 ± 0.46 | 26.24 ± 2.47 | 18.81 ± 1.13 |
| Relation | 40.33 ± 0.77 | 32.92 ± 1.34 | 36.62 ± 0.92 | 23.03 ± 1.11 | 20.99 ± 3.11 | 23.60 ± 0.82 | 34.46 ± 1.30 | 25.52 ± 1.36 |
| Relation+UM | 47.41 ± 4.96 | 39.67 ± 3.33 | 43.54 ± 5.38 | 26.97 ± 1.47 | 17.98 ± 1.24 | 24.23 ± 1.11 | 38.10 ± 1.11 | 26.82 ± 1.24 |
| Relation+ours | 41.67 ± 1.02 | 34.88 ± 1.02 | 38.28 ± 0.99 | 19.73 ± 2.91 | 15.29 ± 2.01 | 23.63 ± 1.30 | 36.96 ± 2.73 | 23.90 ± 0.97 |
| Fdbd | 38.90 ± 0.79 | 29.54 ± 1.28 | 34.22 ± 0.79 | 19.42 ± 0.43 | 20.95 ± 3.51 | 21.28 ± 0.93 | 27.94 ± 0.73 | 22.40 ± 1.30 |
| Fdbd+UM | 40.39 ± 0.43 | 35.01 ± 1.13 | 37.70 ± 2.06 | 21.43 ± 4.46 | 17.81 ± 2.95 | 25.12 ± 2.84 | 32.84 ± 1.89 | 24.30 ± 0.93 |
| Fdbd+ours | 36.64 ± 0.36 | 28.62 ± 1.16 | 32.63 ± 0.60 | 17.82 ± 2.25 | 15.76 ± 2.82 | 20.21 ± 1.19 | 26.59 ± 0.74 | 20.09 ± 0.84 |
| Nci | 52.94 ± 1.89 | 41.21 ± 0.55 | 47.07 ± 0.99 | 33.39 ± 2.18 | 23.97 ± 3.63 | 23.78 ± 0.32 | 36.28 ± 1.05 | 29.35 ± 1.56 |
| Nci+UM | 49.84 ± 3.06 | 43.58 ± 3.59 | 46.71 ± 3.31 | 32.19 ± 1.69 | 17.26 ± 2.06 | 25.36 ± 1.30 | 45.03 ± 4.49 | 29.96 ± 1.09 |
| Nci+ours | 42.32 ± 1.08 | 36.20 ± 1.76 | 39.26 ± 1.40 | 22.50 ± 3.99 | 16.02 ± 1.32 | 22.41 ± 1.23 | 35.12 ± 1.28 | 24.01 ± 0.92 |

the OOD-supervised signals are used in the training process, training-time methods can be categorized into OOD-free and OOD-needed methods. The representatives of OOD-free methods are (Wei et al., 2022a; Lin et al., 2021). Wei et al. (2022a) decoupled the influence of logits' norm from the training procedure by incorporating LogitNorm into the cross-entropy loss. Lin et al. (2021) exploited intermediate classifier outputs for dynamic and efficient OOD inference. The OOD-needed methods aim to calibrate the model by OOD-supervised signals (Ming et al., 2022a; Katz-Samuels et al., 2022; Du et al., 2022).

### A.1.2 SAMPLE SELECTION

Sample selection has emerged as a powerful technique for enhancing the efficiency and robustness of deep learning model training. Current approaches typically prioritize instances based on their informativeness (Alain et al., 2015), uniqueness (Shi et al., 2021), or confidence levels (Khim et al., 2020), though these methods often incur significant computational overhead. Existing selection strategies can be broadly categorized into two paradigms: (1) static selection methods like Data Pruning (Killamsetty et al., 2021b) and Core Set (Xia et al., 2023), which identify representative subsets before training, and (2) dynamic approaches such as Dynamic Data Pruning (Qin et al., 2023) and Importance Sampling (Jiang et al., 2019), which continuously adjust sample selection during training. Various metrics have been proposed to quantify sample informativeness, including gradient norms (Killamsetty et al., 2021a), loss values (Mindermann et al., 2022), and prediction uncertainty (Chang et al., 2017). While our method UARB shares the fundamental objective of optimizing training through selective instance participation, our method introduces a novel uncertainty-driven selection criterion based on whether the model has "mastered" a given instance. This key distinction enables UARB to automatically adapt the training subset composition without requiring predefined schedules or fixed removal rates, offering significant advantages over conventional approaches. Different from the existing sample selection techniques have primarily focused on classification tasks, UARB is dedicated to enhancing OOD detection. Additionally, we develop an adaptive strategy by incorporating class-informed statistics to determine when mastering an instance.

### A.2 USE OF LLMS

We only sought assistance from LLMs for language polishing.

### A.3 METHOD

#### A.3.1 DETAILED DISCUSSION ON UNCERTAINTY METRICS

The core of our mastery criterion is the principle that a mastered instance should exhibit low and stable uncertainty. This is formalized by requiring both the uncertainty value $U_i(w^{(t)})$ and the

Table 10: Comparison of OOD detection methods (with our proposed UARB) on MedMNIST benchmark.

| Method | MedMNIST | | | |
| --- | --- | --- | --- | --- |
| | Near-OOD | | Far-OOD | |
| | FPR95↓ | AUROC↑ | FPR95↓ | AUROC↑ |
| MSP | 50.38 | 83.08 | 33.91 | 89.23 |
| MSP+ours | 42.07 | 86.63 | 25.57 | 91.28 |
| Energy | 59.91 | 79.76 | 45.4 | 84.06 |
| Energy+ours | 56.28 | 83.75 | 41.57 | 86.31 |
| KNN | 35.82 | 86.84 | 14.57 | 93.23 |
| KNN+ours | 34.47 | 87.74 | 15.23 | 93.49 |
| ReAct | 68.2 | 74.67 | 51.34 | 82.16 |
| ReAct+ours | 52.13 | 83.58 | 38.49 | 87.15 |
| Relation | 51.49 | 79.53 | 27.25 | 89.45 |
| Relation+ours | 49.02 | 83.42 | 27.88 | 90.73 |
| Fdbd | 47.68 | 82.86 | 22.17 | 91.87 |
| Fdbd+ours | 39.35 | 86.06 | 22.18 | 92.54 |
| Nci | 72.1 | 69.65 | 46.37 | 81.43 |
| Nci+ours | 63.35 | 76.82 | 45.66 | 84.65 |

absolute value of its second-order difference $|\Delta^2 U_i(w^{(t)})|$ to fall below small, positive thresholds ($\delta_1$ and $\delta_2$). The choice of the uncertainty metric $U$ directly influences the sensitivity and reliability of this criterion.

We empirically evaluated three candidate metrics for $U$:

- Negative Maximum Softmax Probability (-MSP): $U = -\max(\text{Softmax}(\cdot))$. MSP is a classic confidence score in OOD detection. Its advantages are directness and computational efficiency. Its dynamics effectively capture the model's transition from uncertainty to certainty during training. A low -MSP value indicates high confidence, directly satisfying the $U_i < \delta_1$ and $|\Delta^2 U_i(w^{(t)})| < \delta_2$ condition for mastered instances.

- Entropy: $U = -\sum p_i \log p_i$. Entropy measures the dispersion of the entire predictive distribution. A low Entropy value indicates high confidence, directly satisfying the $U_i < \delta_1$ and $|\Delta^2 U_i(w^{(t)})| < \delta_2$ condition for mastered instances.

- Loss (e.g., Cross-Entropy): The loss function is a primary driver of learning. A low loss value indicates high confidence, directly satisfying the $U_i < \delta_1$ and $|\Delta^2 U_i(w^{(t)})| < \delta_2$ condition for mastered instances.

As shown in Table 8 of the paper, we conducted experiments using these metrics within our UARB framework on the CIFAR-10 benchmark to analyze how the mastery criterion behaves under each metric. The results demonstrate that employing -MSP as the uncertainty score yields the most effective OOD detection performance, achieving the lowest average FPR95 in both near-OOD and far-OOD scenarios. While using Loss or Entropy still provides significant improvements over the baseline, their performance is slightly inferior to -MSP. This empirically confirms that -MSP offers the most reliable and stable signal for our mastery criterion.

### A.3.2 DETAILED DISCUSSION ABOUT THE STABILITY AND NOISE RESILIENCE OF THE SECOND-ORDER DIFFERENCE

The second-order difference is designed to be stable by capturing the acceleration of uncertainty convergence rather than momentary fluctuations. By evaluating uncertainty changes across three consecutive epochs (t, t-1, t-2), the second-order difference inherently smooths out short-term noise and transient variations in training dynamics. This multi-epoch perspective reduces sensitivity to isolated anomalies or batch-specific noise. The second-order difference emphasizes the rate of change in uncertainty decay. When this value approaches zero, it indicates that uncertainty has stabilized around a plateau, signaling instance mastery. This trend-based approach is less affected by absolute noise levels in individual uncertainty estimates.

Unlike zero-order uncertainty (raw values), the second-order difference acts as a high-pass filter that attenuates low-frequency noise while highlighting consistent convergence patterns. This makes it particularly effective in distinguishing true learning signals from stochastic fluctuations. Our mastery criterion requires both the uncertainty itself and its second-order difference to fall below thresholds ($\delta_1$ and $\delta_2$). This dual requirement ensures that only instances with persistently low and stable uncertainty are classified as mastered, reducing false positives from noisy estimates.

## A.4 DETAILED RESULTS

### A.4.1 RESULTS ON MEDMNIST

To further validate the effectiveness of the method, we also conducted experiments on medical datasets. The in-distribution (ID) dataset is OrganAMNIST, the near-OOD datasets include OrganCMNIST, OrganSMNIST, ChestMNIST, and PneumoniaMNIST, while the far-OOD datasets comprise PathMNIST, DermaMNIST, RetinaMNIST, and BloodMNIST. We report the average FPR95 and AUROC for the corresponding scenarios. The results are shown in Table 10. The experimental results demonstrate that the OOD detection performance of the baseline methods improved after integrating our approach, in both near-OOD and far-OOD scenarios, which validates the effectiveness and generalizability of our method.

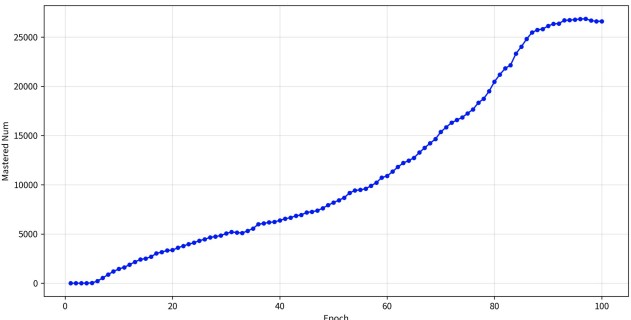

Figure 2: Analysis on the number of instances that meet the mastered criteria.

### A.4.2 ANALYSIS ON THE NUMBER OF INSTANCES THAT MEET THE MASTERED CRITERIA

We have added a figure that explicitly demonstrates the number of instances that meet the mastered criteria. Fig. 2 illustrates the number of mastered instances throughout the training process on the CIFAR-10 dataset. Through it, we can observe that the number of mastered instances increases slowly at first, then rises rapidly, and finally plateaus.

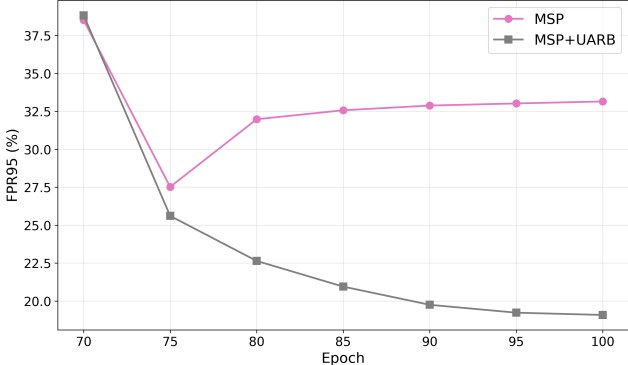

Figure 3: Analysis on the number of instances that meet the mastered criteria.

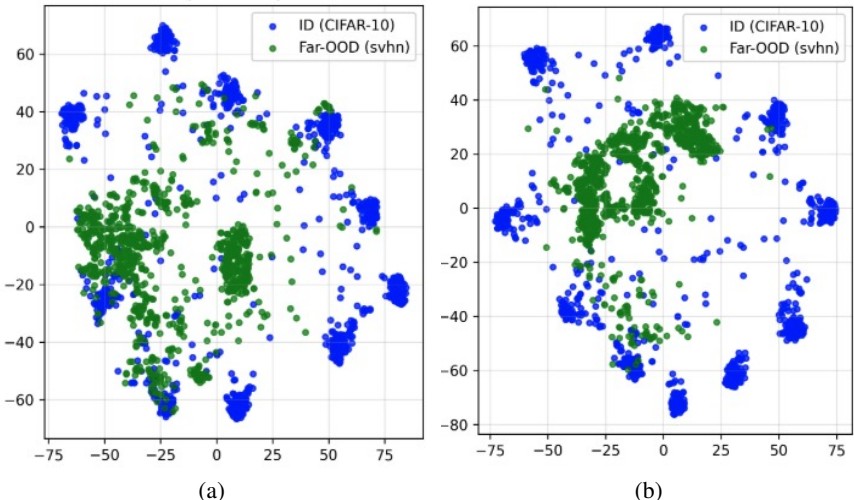

Figure 4: T-SNE about the baseline method and our approach.

### A.4.3 INVESTIGATE WHETHER UARB MITIGATES THE OVERFITTING

To directly investigate whether UARB mitigates the overfitting in later training stages, we conducted a systematic evaluation comparing the OOD detection performance of baseline methods versus their UARB-enhanced versions across different training phases. Fig. 3 demonstrates the average FPR95 of the baseline and the UARB-enhanced version at different later stages of training. The results reveals that UARB significantly alleviates the overfitting during later training stages.

Furthermore, we visualized the t-SNE plots comparing the baseline method (Fig. 4(a)) and our enhanced method (Fig. 4(b)). The results clearly demonstrate that our method achieves superior separation between in-distribution (ID) and out-of-distribution (OOD) data. This observation provides visual confirmation that overfitting has been effectively mitigated to a certain extent.

Table 11: Comparison (AUROC↑) on CIFAR-10 benchmark. All values are percentages.

| Method | CIFAR-100 | TIN | Near-Avg | MNIST | SVHM | Texture | Places365 | Far-Avg |
|---|---|---|---|---|---|---|---|---|
| MSP | 87.25 ± 0.34 | 89.10 ± 0.38 | 88.18 ± 0.34 | 91.88 ± 0.39 | 91.92 ± 1.12 | 90.94 ± 0.19 | 89.28 ± 0.35 | 91.01 ± 0.47 |
| MSP+UM | 86.97 ± 0.30 | 89.00 ± 0.60 | 87.98 ± 0.45 | 89.90 ± 3.59 | 95.89 ± 1.23 | 91.76 ± 0.62 | 89.62 ± 1.05 | 91.79 ± 0.48 |
| MSP+ours | 88.55 ± 0.21 | 90.27 ± 0.30 | 89.41 ± 0.25 | 93.85 ± 0.81 | 95.30 ± 1.26 | 92.24 ± 0.09 | 90.54 ± 0.45 | 92.98 ± 0.36 |
| Energy | 86.24 ± 0.48 | 88.95 ± 0.50 | 87.60 ± 0.47 | 93.51 ± 0.59 | 91.54 ± 1.91 | 90.78 ± 0.34 | 89.75 ± 0.41 | 91.40 ± 0.75 |
| Energy+UM | 86.89 ± 0.36 | 90.28 ± 0.56 | 88.59 ± 0.33 | 93.83 ± 2.11 | 95.70 ± 1.90 | 91.91 ± 1.08 | 92.94 ± 1.09 | 93.60 ± 0.36 |
| Energy+ours | 89.73 ± 0.05 | 92.12 ± 0.22 | 90.93 ± 0.13 | 97.22 ± 0.45 | 97.17 ± 1.02 | 93.75 ± 0.21 | 93.11 ± 0.52 | 95.31 ± 0.23 |
| KNN | 89.78 ± 0.30 | 91.76 ± 0.28 | 90.77 ± 0.28 | 93.95 ± 0.12 | 93.52 ± 0.83 | 93.65 ± 0.07 | 92.14 ± 0.32 | 93.31 ± 0.29 |
| KNN+UM | 86.61 ± 0.54 | 88.34 ± 1.09 | 87.48 ± 0.82 | 94.64 ± 0.24 | 90.76 ± 1.52 | 89.83 ± 1.05 | 90.79 ± 0.78 | 91.50 ± 0.30 |
| KNN+ours | 89.19 ± 0.13 | 91.22 ± 0.17 | 90.20 ± 0.14 | 95.66 ± 1.06 | 93.83 ± 1.76 | 93.55 ± 0.14 | 92.37 ± 0.58 | 93.85 ± 0.63 |
| ReAct | 86.63 ± 0.54 | 89.18 ± 0.30 | 87.90 ± 0.42 | 92.30 ± 1.02 | 91.42 ± 1.45 | 91.42 ± 0.64 | 90.45 ± 0.45 | 91.40 ± 0.68 |
| ReAct+UM | 86.72 ± 0.89 | 90.02 ± 0.57 | 88.37 ± 0.69 | 92.80 ± 1.76 | 95.14 ± 2.39 | 92.38 ± 1.35 | 93.11 ± 1.01 | 93.36 ± 0.70 |
| ReAct+ours | 89.66 ± 0.01 | 92.02 ± 0.19 | 90.84 ± 0.09 | 96.92 ± 0.66 | 96.90 ± 1.25 | 93.86 ± 0.07 | 93.22 ± 0.84 | 95.22 ± 0.34 |
| Relation | 89.01 ± 0.34 | 90.94 ± 0.31 | 89.97 ± 0.31 | 93.26 ± 0.25 | 93.31 ± 1.06 | 92.90 ± 0.09 | 90.78 ± 0.38 | 92.56 ± 0.40 |
| Relation+UM | 86.27 ± 0.49 | 88.85 ± 0.70 | 87.56 ± 0.79 | 91.39 ± 0.42 | 94.73 ± 0.25 | 92.62 ± 0.32 | 88.78 ± 0.21 | 91.88 ± 0.19 |
| Relation+ours | 88.05 ± 0.54 | 89.91 ± 0.64 | 88.98 ± 0.59 | 94.01 ± 0.61 | 95.18 ± 1.30 | 92.72 ± 0.64 | 90.04 ± 1.06 | 92.99 ± 0.08 |
| Fdbd | 89.59 ± 0.25 | 91.77 ± 0.29 | 90.68 ± 0.26 | 94.46 ± 0.12 | 93.32 ± 1.13 | 93.73 ± 0.13 | 92.21 ± 0.23 | 93.43 ± 0.38 |
| Fdbd+UM | 87.67 ± 0.89 | 91.09 ± 0.57 | 89.38 ± 0.51 | 92.62 ± 1.72 | 95.08 ± 1.56 | 92.35 ± 1.44 | 91.79 ± 1.01 | 93.21 ± 0.22 |
| Fdbd+ours | 89.52 ± 0.14 | 91.96 ± 0.24 | 90.74 ± 0.19 | 95.16 ± 0.71 | 95.09 ± 1.90 | 94.20 ± 0.32 | 92.84 ± 0.31 | 94.32 ± 0.52 |
| Nci | 87.81 ± 0.37 | 89.69 ± 0.22 | 88.75 ± 0.26 | 90.96 ± 0.26 | 92.27 ± 1.05 | 92.68 ± 0.16 | 90.30 ± 0.31 | 91.55 ± 0.40 |
| Nci+UM | 86.33 ± 0.63 | 88.16 ± 0.88 | 87.24 ± 0.76 | 91.19 ± 0.62 | 94.88 ± 0.85 | 92.68 ± 0.33 | 87.96 ± 1.10 | 91.68 ± 0.14 |
| Nci+ours | 88.09 ± 0.29 | 89.88 ± 0.38 | 88.98 ± 0.33 9 | 3.58 ± 1.23 | 95.52 ± 1.10 | 93.52 ± 0.34 | 90.28 ± 0.32 | 93.22 ± 0.42 |

### A.4.4 DETAILED RESULTS ON CIFAR

Tables 9 and Table 11 present the detailed FPR95 and AUROC results of our method and baseline methods on CIFAR-10. Although performance varies across different OOD datasets, our method demonstrates nearly comprehensive superiority over baselines in terms of average performance.

Furthermore, the proposed approach exhibits consistent effectiveness in both Near-OOD and Far-OOD scenarios. When combined with various baselines, our method also shows orthogonality, indicating its compatibility and complementary advantages.

Tables 12 and Table 13 present the detailed FPR95 and AUROC results of our method and baseline methods on CIFAR-100. The proposed approach exhibits consistent effectiveness in both Near-OOD and Far-OOD scenarios. When combined with various baselines, our method also shows orthogonality.

Table 12: Comparison (FPR95↓) on CIFAR-100 benchmark. All values are percentages.

| Method | CIFAR-10 | TIN | Near-Avg | MNIST | SVHN | Texture | Places365 | Far-Avg |
|---|---|---|---|---|---|---|---|---|
| MSP | 59.70 ± 0.73 | 50.60 ± 0.95 | 55.15 ± 0.13 | 59.49 ± 1.46 | 56.51 ± 5.82 | 61.59 ± 0.85 | 56.70 ± 0.75 | 58.57 ± 1.50 |
| MSP+UM | 59.35 ± 0.58 | 51.19 ± 0.44 | 55.27 ± 0.24 | 56.21 ± 2.12 | 55.02 ± 2.49 | 60.87 ± 2.67 | 57.78 ± 0.21 | 57.47 ± 1.14 |
| MSP+ours | 59.13 ± 0.26 | 50.87 ± 0.56 | 55.00 ± 0.34 | 54.85 ± 1.31 | 50.32 ± 7.88 | 60.22 ± 0.45 | 55.82 ± 0.41 | 55.30 ± 2.17 |
| Energy | 60.21 ± 0.81 | 51.77 ± 0.53 | 55.99 ± 0.27 | 55.13 ± 1.33 | 49.04 ± 7.13 | 63.00 ± 0.29 | 56.92 ± 0.82 | 56.02 ± 1.58 |
| Energy+UM | 60.65 ± 1.21 | 54.07 ± 0.44 | 57.36 ± 0.48 | 51.96 ± 2.10 | 47.26 ± 1.17 | 63.49 ± 3.47 | 59.85 ± 0.19 | 55.64 ± 1.09 |
| Energy+ours | 59.50 ± 0.47 | 51.59 ± 0.65 | 55.54 ± 0.54 | 51.65 ± 1.62 | 43.20 ± 9.54 | 60.32 ± 0.69 | 57.02 ± 0.67 | 53.05 ± 2.70 |
| KNN | 73.70 ± 1.92 | 49.47 ± 0.37 | 61.59 ± 0.83 | 50.81 ± 1.44 | 56.54 ± 10.90 | 52.73 ± 1.86 | 59.99 ± 0.59 | 55.02 ± 3.43 |
| KNN+UM | 77.03 ± 0.78 | 51.29 ± 0.55 | 64.16 ± 0.46 | 43.34 ± 3.76 | 47.75 ± 4.16 | 51.06 ± 0.39 | 64.00 ± 0.35 | 51.54 ± 2.08 |
| KNN+ours | 72.09 ± 2.04 | 49.73 ± 0.36 | 60.91 ± 1.18 | 46.46 ± 3.99 | 46.32 ± 7.30 | 53.40 ± 2.27 | 60.02 ± 0.87 | 51.55 ± 2.94 |
| ReAct | 61.48 ± 1.16 | 51.70 ± 0.57 | 56.59 ± 0.31 | 57.66 ± 1.46 | 45.78 ± 5.58 | 56.34 ± 0.83 | 55.34 ± 0.88 | 53.78 ± 1.18 |
| ReAct+UM | 62.80 ± 0.91 | 53.84 ± 0.50 | 58.32 ± 0.21 | 55.66 ± 1.36 | 46.37 ± 1.79 | 55.33 ± 2.08 | 59.04 ± 0.55 | 54.10 ± 0.57 |
| ReAct+ours | 61.17 ± 0.57 | 51.61 ± 0.76 | 56.39 ± 0.54 | 53.34 ± 1.78 | 39.90 ± 8.02 | 52.87 ± 0.58 | 55.26 ± 0.93 | 50.34 ± 2.38 |
| Relation | 72.03 ± 2.55 | 49.56 ± 0.59 | 60.80 ± 0.99 | 52.46 ± 1.97 | 57.81 ± 9.79 | 54.34 ± 2.29 | 61.57 ± 0.75 | 56.55 ± 3.25 |
| Relation+UM | 72.16 ± 0.75 | 50.13 ± 0.16 | 61.14 ± 0.30 | 46.34 ± 2.01 | 49.38 ± 3.92 | 52.30 ± 0.81 | 62.60 ± 0.42 | 52.66 ± 1.63 |
| Relation+ours | 71.62 ± 1.48 | 50.17 ± 0.51 | 60.89 ± 0.94 | 48.41 ± 3.71 | 47.37 ± 8.66 | 54.55 ± 1.69 | 60.77 ± 1.19 | 52.77 ± 3.20 |
| Fdbd | 65.41 ± 1.42 | 47.81 ± 0.66 | 56.61 ± 0.73 | 55.02 ± 1.35 | 54.85 ± 7.13 | 53.64 ± 1.47 | 57.19 ± 0.68 | 55.18 ± 2.12 |
| Fdbd+UM | 66.76 ± 0.86 | 48.67 ± 0.24 | 57.72 ± 0.55 | 47.22 ± 1.00 | 48.51 ± 3.38 | 52.05 ± 1.47 | 58.62 ± 0.24 | 51.60 ± 1.04 |
| Fdbd+ours | 64.46 ± 1.20 | 48.33 ± 0.66 | 56.39 ± 0.81 | 49.37 ± 3.07 | 46.78 ± 7.62 | 52.84 ± 1.77 | 56.26 ± 0.17 | 51.31 ± 2.83 |
| Nci | 63.47 ± 1.47 | 48.06 ± 0.79 | 55.76 ± 0.60 | 54.74 ± 2.24 | 49.47 ± 5.89 | 47.71 ± 0.91 | 53.83 ± 0.30 | 51.44 ± 1.52 |
| Nci+UM | 63.84 ± 0.40 | 49.34 ± 0.34 | 56.59 ± 0.18 | 47.64 ± 3.48 | 45.04 ± 0.97 | 46.00 ± 1.06 | 55.09 ± 0.41 | 48.44 ± 0.97 |
| Nci+ours | 63.96 ± 0.88 | 49.51 ± 0.75 | 56.74 ± 0.28 | 49.71 ± 2.39 | 43.53 ± 5.67 | 47.29 ± 1.03 | 53.17 ± 0.21 | 48.43 ± 1.96 |

Table 13: Comparison (AUROC↑) on CIFAR-100 benchmark. All values are percentages.

| Method | CIFAR-10 | TIN | Near-Avg | MNIST | SVHN | Texture | Places365 | Far-Avg |
|---|---|---|---|---|---|---|---|---|
| MSP | 78.37 ± 0.20 | 82.16 ± 0.21 | 80.27 ± 0.11 | 75.55 ± 0.50 | 79.97 ± 2.60 | 77.34 ± 0.78 | 79.37 ± 0.35 | 78.06 ± 0.88 |
| MSP+UM | 78.29 ± 0.11 | 82.07 ± 0.26 | 80.18 ± 0.16 | 77.53 ± 1.54 | 80.46 ± 0.84 | 78.05 ± 0.90 | 78.99 ± 0.18 | 78.76 ± 0.50 |
| MSP+ours | 78.64 ± 0.06 | 82.25 ± 0.13 | 80.45 ± 0.09 | 76.93 ± 0.22 | 82.42 ± 2.58 | 78.04 ± 0.40 | 79.62 ± 0.26 | 79.25 ± 0.71 |
| Energy | 78.95 ± 0.23 | 82.80 ± 0.12 | 80.88 ± 0.08 | 78.55 ± 0.32 | 84.11 ± 2.92 | 77.92 ± 0.89 | 79.73 ± 0.56 | 80.08 ± 0.99 |
| Energy+UM | 78.48 ± 0.36 | 82.22 ± 0.17 | 80.35 ± 0.21 | 79.90 ± 1.75 | 83.81 ± 0.12 | 78.56 ± 1.25 | 78.84 ± 0.15 | 80.28 ± 0.76 |
| Energy+ours | 79.25 ± 0.15 | 82.87 ± 0.20 | 81.06 ± 0.17 | 79.35 ± 1.15 | 85.93 ± 3.07 | 79.08 ± 0.76 | 79.90 ± 0.26 | 81.06 ± 1.08 |
| KNN | 76.97 ± 0.33 | 83.56 ± 0.13 | 80.26 ± 0.10 | 82.26 ± 1.26 | 83.60 ± 2.95 | 84.04 ± 0.24 | 79.60 ± 0.12 | 82.38 ± 1.10 |
| KNN +UM | 76.46 ± 0.35 | 83.16 ± 0.12 | 79.81 ± 0.23 | 83.95 ± 2.38 | 86.22 ± 0.73 | 84.72 ± 0.24 | 78.54 ± 0.18 | 83.36 ± 0.84 |
| KNN+ours | 77.26 ± 0.26 | 83.56 ± 0.05 | 80.41 ± 0.13 | 82.29 ± 1.66 | 85.45 ± 1.88 | 84.07 ± 0.60 | 79.84 ± 0.20 | 82.91 ± 1.01 |
| ReAct | 78.71 ± 0.27 | 82.88 ± 0.15 | 80.80 ± 0.11 | 77.84 ± 0.27 | 84.92 ± 1.95 | 79.66 ± 0.84 | 80.18 ± 0.61 | 80.65 ± 0.72 |
| ReAct+UM | 78.00 ± 0.30 | 82.31 ± 0.08 | 80.15 ± 0.17 | 79.07 ± 1.62 | 84.19 ± 0.33 | 80.67 ± 0.78 | 79.14 ± 0.21 | 80.77 ± 0.55 |
| ReAct+ours | 78.94 ± 0.15 | 82.93 ± 0.21 | 80.93 ± 0.17 | 78.88 ± 1.19 | 86.55 ± 2.79 | 80.89 ± 0.69 | 80.30 ± 0.28 | 81.65 ± 1.04 |
| Relation | 77.75 ± 0.35 | 83.66 ± 0.11 | 80.71 ± 0.13 | 79.94 ± 0.67 | 82.81 ± 2.48 | 81.29 ± 0.66 | 79.73 ± 0.36 | 80.94 ± 0.96 |
| Relation+UM | 77.67 ± 0.04 | 83.41 ± 0.12 | 80.54 ± 0.05 | 81.54 ± 1.29 | 84.33 ± 0.61 | 82.10 ± 0.41 | 79.18 ± 0.05 | 81.79 ± 0.50 |
| Relation+ours | 78.02 ± 0.14 | 83.65 ± 0.05 | 80.83 ± 0.06 | 80.39 ± 0.98 | 85.10 ± 2.57 | 81.69 ± 0.37 | 79.97 ± 0.27 | 81.79 ± 0.98 |
| Fdbd | 78.10 ± 0.23 | 84.05 ± 0.11 | 81.07 ± 0.09 | 78.59 ± 0.85 | 81.03 ± 2.36 | 81.35 ± 0.65 | 79.94 ± 0.37 | 80.23 ± 0.91 |
| Fdbd+UM | 77.74 ± 0.07 | 83.87 ± 0.17 | 80.80 ± 0.11 | 81.16 ± 0.84 | 82.64 ± 0.99 | 81.70 ± 0.54 | 79.49 ± 0.09 | 81.25 ± 0.40 |
| Fdbd+ours | 78.40 ± 0.15 | 84.11 ± 0.08 | 81.25 ± 0.10 | 79.67 ± 1.57 | 83.43 ± 2.13 | 81.60 ± 0.46 | 80.32 ± 0.14 | 81.26 ± 1.06 |
| Nci | 78.34 ± 0.24 | 83.65 ± 0.15 | 81.00 ± 0.05 | 79.81 ± 0.24 | 83.21 ± 1.91 | 83.87 ± 0.41 | 80.95 ± 0.22 | 81.96 ± 0.63 |
| Nci+UM | 78.19 ± 0.19 | 83.54 ± 0.18 | 80.87 ± 0.18 | 81.18 ± 2.16 | 84.73 ± 0.09 | 84.53 ± 0.32 | 80.46 ± 0.11 | 82.73 ± 0.55 |
| Nci+ours | 78.47 ± 0.01 | 83.73 ± 0.04 | 81.10 ± 0.02 | 79.88 ± 0.49 | 84.73 ± 1.68 | 84.10 ± 0.31 | 81.30 ± 0.20 | 82.50 ± 0.63 |

### A.4.5 Detailed Results on ImageNet

Table 14 reports the detailed AUROC results of our method and the baseline methods on ImageNet. The results demonstrate that our method achieves near-comprehensive superiority over the baselines. Furthermore, it exhibits consistent effectiveness in both Near-OOD and Far-OOD scenarios. When combined with different baselines, our method also demonstrates orthogonality, highlighting its compatibility and complementary advantages.

Table 14: Comparison (AUROC↑) on ImageNet benchmark. All values are percentages.

| Method | Ssb_hard | Ninco | Near-Avg | iNaturalist | Texture | Openimage-o | Far-Avg |
|---|---|---|---|---|---|---|---|
| MSP | 89.52 ± 0.08 | 90.77 ± 0.04 | 90.15 ± 0.04 | 94.11 ± 0.13 | 93.64 ± 0.12 | 92.74 ± 0.06 | 93.50 ± 0.07 |
| MSP+UM | 89.53 ± 0.05 | 90.63 ± 0.03 | 90.08 ± 0.02 | 94.16 ± 0.23 | 93.78 ± 0.15 | 92.65 ± 0.12 | 93.53 ± 0.17 |
| MSP+ours | 89.85 ± 0.10 | 91.02 ± 0.16 | 90.43 ± 0.12 | 94.44 ± 0.15 | 93.91 ± 0.20 | 93.00 ± 0.18 | 93.78 ± 0.17 |
| Energy | 90.12 ± 0.08 | 91.48 ± 0.12 | 90.80 ± 0.06 | 95.39 ± 0.25 | 94.91 ± 0.12 | 94.29 ± 0.05 | 94.87 ± 0.07 |
| Energy+UM | 89.40 ± 0.13 | 90.64 ± 0.15 | 90.02 ± 0.10 | 94.82 ± 0.30 | 94.59 ± 0.32 | 93.57 ± 0.33 | 94.33 ± 0.30 |
| Energy+ours | 89.40 ± 0.13 | 90.64 ± 0.15 | 90.02 ± 0.10 | 94.82 ± 0.30 | 94.59 ± 0.32 | 93.57 ± 0.33 | 94.33 ± 0.30 |
| KNN | 90.88 ± 0.09 | 93.12 ± 0.13 | 92.00 ± 0.10 | 95.97 ± 0.11 | 97.97 ± 0.08 | 95.51 ± 0.12 | 96.48 ± 0.09 |
| KNN+UM | 90.78 ± 0.22 | 92.89 ± 0.17 | 91.84 ± 0.13 | 95.76 ± 0.31 | 98.01 ± 0.07 | 95.43 ± 0.13 | 96.40 ± 0.14 |
| KNN+ours | 91.28 ± 0.15 | 93.22 ± 0.14 | 92.25 ± 0.10 | 96.21 ± 0.06 | 98.03 ± 0.02 | 95.71 ± 0.01 | 96.65 ± 0.01 |
| ReAct | 90.05 ± 0.14 | 91.69 ± 0.13 | 90.87 ± 0.11 | 95.52 ± 0.26 | 95.34 ± 0.09 | 94.97 ± 0.05 | 95.12 ± 0.12 |
| ReAct+UM | 89.34 ± 0.21 | 90.91 ± 0.14 | 90.12 ± 0.10 | 95.10 ± 0.30 | 95.14 ± 0.23 | 93.92 ± 0.33 | 94.72 ± 0.27 |
| ReAct+ours | 90.19 ± 0.12 | 91.72 ± 0.17 | 90.95 ± 0.03 | 95.55 ± 0.21 | 95.48 ± 0.12 | 94.40 ± 0.16 | 95.14 ± 0.13 |
| Relation | 91.32 ± 0.06 | 93.17 ± 0.05 | 92.25 ± 0.05 | 96.20 ± 0.09 | 97.00 ± 0.08 | 95.29 ± 0.06 | 96.16 ± 0.08 |
| Relation +UM | 91.24 ± 0.10 | 92.99 ± 0.18 | 92.11 ± 0.13 | 96.14 ± 0.11 | 96.99 ± 0.03 | 95.17 ± 0.05 | 96.10 ± 0.05 |
| Relation +ours | 91.50 ± 0.07 | 93.21 ± 0.16 | 92.36 ± 0.10 | 96.35 ± 0.05 | 97.02 ± 0.02 | 95.37 ± 0.06 | 96.25 ± 0.04 |
| Fdbd | 90.90 ± 0.10 | 93.00 ± 0.07 | 91.95 ± 0.06 | 96.16 ± 0.04 | 96.89 ± 0.11 | 95.20 ± 0.05 | 96.08 ± 0.06 |
| Fdbd+UM | 91.04 ± 0.07 | 93.11 ± 0.09 | 92.07 ± 0.02 | 96.38 ± 0.11 | 97.00 ± 0.06 | 95.36 ± 0.04 | 96.25 ± 0.05 |
| Fdbd+ours | 91.23 ± 0.14 | 93.31 ± 0.10 | 92.27 ± 0.08 | 96.51 ± 0.02 | 97.08 ± 0.04 | 95.49 ± 0.05 | 96.36 ± 0.04 |
| Nci | 90.81 ± 0.03 | 92.49 ± 0.08 | 91.65 ± 0.03 | 95.82 ± 0.03 | 96.94 ± 0.09 | 94.97 ± 0.05 | 95.91 ± 0.05 |
| Nci+UM | 90.90 ± 0.09 | 92.53 ± 0.06 | 91.71 ± 0.02 | 96.04 ± 0.09 | 97.03 ± 0.06 | 95.10 ± 0.05 | 96.06 ± 0.05 |
| Nci+ ours | 91.14 ± 0.06 | 92.72 ± 0.10 | 91.93 ± 0.08 | 96.16 ± 0.05 | 97.06 ± 0.08 | 95.21 ± 0.07 | 96.14 ± 0.07 |

Table 15: Comparison (FPR95↓) on CIFAR-10 benchmark with ResNet-50. All values are percentages.

| Method | CIFAR-100 | TIN | Near-Avg | MNIST | SVHM | Texture | Places365 | Far-Avg |
|---|---|---|---|---|---|---|---|---|
| MSP | 51.35 ± 1.58 | 47.70 ± 1.65 | 49.52 ± 1.61 | 42.02 ± 4.93 | 44.27 ± 5.56 | 48.39 ± 1.82 | 46.54 ± 1.53 | 45.30 ± 1.91 |
| MSP+ours | 47.14 ± 0.15 | 43.19 ± 0.04 | 45.17 ± 0.05 | 39.10 ± 4.89 | 35.09 ± 4.16 | 43.47 ± 0.51 | 43.21 ± 0.30 | 40.22 ± 1.83 |
| Energy | 54.42 ± 2.60 | 48.58 ± 2.78 | 51.50 ± 2.68 | 36.52 ± 5.20 | 52.06 ± 8.05 | 49.25 ± 3.56 | 44.47 ± 3.50 | 45.58 ± 4.48 |
| Energy+ours | 47.16 ± 0.46 | 41.31 ± 0.45 | 44.24 ± 0.43 | 29.10 ± 6.57 | 39.81 ± 4.30 | 44.09 ± 0.54 | 38.31 ± 0.80 | 37.83 ± 2.26 |
| KNN | 51.04 ± 1.41 | 45.59 ± 0.51 | 48.31 ± 0.87 | 33.75 ± 6.81 | 45.65 ± 2.99 | 44.39 ± 1.10 | 43.47 ± 1.32 | 41.81 ± 1.35 |
| KNN+ours | 48.99 ± 0.85 | 43.11 ± 1.02 | 46.05 ± 0.93 | 31.77 ± 3.30 | 38.54 ± 2.02 | 43.40 ± 1.90 | 42.36 ± 0.29 | 39.02 ± 0.64 |
| ReAct | 66.70 ± 0.56 | 64.43 ± 1.22 | 65.57 ± 0.68 | 68.85 ± 3.06 | 69.58 ± 7.90 | 66.94 ± 3.06 | 56.39 ± 0.47 | 65.44 ± 2.36 |
| ReAct+ours | 61.04 ± 2.12 | 57.70 ± 2.72 | 59.37 ± 2.42 | 56.82 ± 8.21 | 59.21 ± 5.76 | 62.12 ± 2.10 | 50.73 ± 3.18 | 57.22 ± 2.93 |
| Relation | 55.91 ± 1.11 | 51.27 ± 2.24 | 53.59 ± 1.62 | 41.28 ± 8.47 | 41.97 ± 3.03 | 48.30 ± 2.86 | 52.61 ± 2.41 | 46.04 ± 0.75 |
| Relation+ours | 57.33 ± 3.02 | 50.91 ± 4.30 | 54.12 ± 3.63 | 35.03 ± 2.05 | 36.13 ± 2.87 | 56.95 ± 9.57 | 54.84 ± 1.41 | 45.74 ± 3.51 |
| Fdbd | 59.47 ± 0.97 | 51.95 ± 0.54 | 55.71 ± 0.73 | 39.62 ± 11.92 | 62.51 ± 4.82 | 52.15 ± 3.36 | 47.49 ± 2.15 | 50.44 ± 2.56 |
| Fdbd+ours | 51.51 ± 1.14 | 45.16 ± 1.30 | 48.33 ± 1.21 | 34.84 ± 4.46 | 53.74 ± 5.92 | 45.03 ± 2.06 | 40.19 ± 0.75 | 43.45 ± 2.19 |
| Nci | 59.41 ± 2.39 | 55.85 ± 2.06 | 57.63 ± 2.23 | 44.96 ± 6.21 | 43.07 ± 6.11 | 58.55 ± 1.92 | 56.54 ± 3.81 | 50.78 ± 1.31 |
| Nci+ours | 59.81 ± 0.99 | 54.38 ± 1.01 | 57.10 ± 0.75 | 45.35 ± 12.11 | 40.91 ± 4.31 | 60.83 ± 5.05 | 56.98 ± 1.35 | 51.02 ± 3.70 |
| DICE | 56.24 ± 3.50 | 49.75 ± 3.92 | 53.00 ± 3.71 | 35.14 ± 10.68 | 48.63 ± 5.52 | 50.28 ± 5.67 | 48.32 ± 5.17 | 45.59 ± 5.82 |
| DICE+ours | 48.77 ± 0.55 | 42.52 ± 0.64 | 45.64 ± 0.57 | 25.03 ± 6.57 | 36.04 ± 6.69 | 46.29 ± 1.94 | 41.40 ± 1.35 | 37.19 ± 2.70 |
| Scale | 55.55 ± 3.95 | 50.65 ± 4.17 | 53.10 ± 4.06 | 45.79 ± 3.73 | 50.97 ± 9.42 | 52.08 ± 6.62 | 47.29 ± 4.75 | 49.03 ± 5.98 |
| Scale+ours | 48.37 ± 0.32 | 42.43 ± 0.41 | 45.40 ± 0.36 | 33.75 ± 6.51 | 39.96 ± 5.20 | 45.68 ± 2.54 | 40.11 ± 1.62 | 39.87 ± 2.18 |

### A.4.6 Computational and Memory Overhead of UARB

We rigorously measured the training time and GPU memory usage for a standard benchmark (CIFAR-10 with a ResNet-18 backbone) to quantify the overhead. The results are as follows: (1) Baseline Method: 2030 seconds, 1452 MiB GPU memory; (2) Our Method (UARB): 2371 seconds, 1456 MiB GPU memory. The additional 341 seconds of training time in UARB is a relatively small cost in the context of a full training run. We adhere to the first-in-first-out principle and utilize a queue to track the uncertainty of each sample, which does not significantly increase memory costs.

### A.4.7 More results on different backbone

To further assess the applicability of UARB, we conducted experiments using ResNet-50 as the backbone. The results in Table 15 demonstrate that integrating UARB consistently improves OOD detection performance across nearly all baseline methods.

### A.4.8 More results about two new baselines

We added two new baselines including DICE Sun & Li (2022) and Scale Xu et al. (2024). The experimental results are as follows. Experimental results in Table 16 and Table 17 demonstrate that

Table 16: Comparison (FPR95↓) on CIFAR-10 benchmark. All values are percentages.

| Method | CIFAR-100 | TIN | Near-Avg | MNIST | SVHM | Texture | Places365 | Far-Avg |
|---|---|---|---|---|---|---|---|---|
| DICE | 76.17 ± 3.28 | 68.56 ± 5.06 | 72.36 ± 4.02 | 21.36 ± 2.55 | 32.11 ± 2.68 | 56.99 ± 4.53 | 81.73 ± 8.72 | 48.05 ± 2.55 |
| DICE+ours | 66.20 ± 2.22 | 58.23 ± 4.46 | 62.21 ± 3.14 | 7.61 ± 1.94 | 9.46 ± 4.37 | 51.78 ± 4.62 | 67.11 ± 13.08 | 33.99 ± 0.96 |
| Scale | 83.86 ± 1.45 | 80.96 ± 1.28 | 82.41 ± 1.37 | 51.44 ± 9.24 | 72.30 ± 12.01 | 86.84 ± 4.39 | 70.61 ± 3.43 | 70.30 ± 5.92 |
| Scale+ours | 62.91 ± 3.33 | 53.93 ± 3.20 | 58.42 ± 3.24 | 18.11 ± 2.82 | 12.40 ± 1.56 | 48.29 ± 11.90 | 42.47 ± 3.67 | 30.32 ± 2.95 |

Table 17: Comparison (AUROC↑) on CIFAR-10 benchmark. All values are percentages.

| Method | CIFAR-100 | TIN | Near-Avg | MNIST | SVHM | Texture | Places365 | Far-Avg |
|---|---|---|---|---|---|---|---|---|
| DICE | 79.17 ± 0.70 | 81.65 ± 1.79 | 80.41 ± 1.19 | 94.43 ± 2.00 | 90.68 ± 1.38 | 84.43 ± 0.53 | 75.88 ± 4.49 | 86.35 ± 1.52 |
| DICE+ours | 81.53 ± 0.67 | 83.19 ± 1.15 | 82.36 ± 0.84 | 98.24 ± 0.51 | 97.27 ± 1.43 | 84.87 ± 1.96 | 80.94 ± 4.13 | 90.33 ± 0.12 |
| Scale | 79.92 ± 0.61 | 82.94 ± 0.42 | 81.43 ± 0.52 | 90.62 ± 1.60 | 82.85 ± 3.83 | 81.16 ± 1.72 | 86.28 ± 1.59 | 85.23 ± 1.69 |
| Scale+ours | 86.11 ± 0.97 | 88.78 ± 0.66 | 87.45 ± 0.80 | 95.22 ± 0.59 | 96.66 ± 0.79 | 90.19 ± 1.95 | 90.99 ± 0.89 | 93.26 ± 0.47 |

both DICE and Scale exhibit significant improvements in FPR95 and AUROC under both near-OOD and far-OOD scenarios after incorporating UARB.

Table 18: Sensitivity analysis of $\delta_1$.

| Method | CIFAR-10 | | | |
|---|---|---|---|---|
| | Near-OOD | | Far-OOD | |
| | FPR95↓ | AUROC↑ | FPR95↓ | AUROC↑ |
| $\delta_1$=0.01 | 33.10 ± 1.35 | 89.69 ± 0.49 | 20.10 ± 0.66 | 93.54 ± 0.40 |
| $\delta_1$=0.005 | 33.57 ± 0.46 | 89.72 ± 0.11 | 21.39 ± 1.01 | 92.97 ± 0.44 |
| $\delta_1$=0.001 | 33.12 ± 1.30 | 89.71 ± 0.21 | 21.42 ± 0.56 | 92.97 ± 0.30 |
| $\delta_1$=0.05 | 31.71 ± 0.79 | 90.01 ± 0.24 | 20.90 ± 1.44 | 93.30 ± 0.69 |
| $\delta_1$=0.1 | 32.71 ± 0.51 | 89.74 ± 0.10 | 20.81 ± 0.63 | 93.02 ± 0.34 |

A.4.9 MORE RESULTS ABOUT SENSITIVITY ANALYSIS OF $\delta_1$ AND $\delta_2$

$\delta_1$ and $\delta_2$ in Eq. 7 denote the positive thresholds for mastered criterion. Table 18 and Table 19 present the results of the sensitivity analysis for parameters $\delta_1$ and $\delta_2$. The results indicate that our method is not particularly sensitive to the selection of $\delta_1$ and $\delta_1$.

A.4.10 MORE RESULTS ABOUT SENSITIVITY ANALYSIS OF $\gamma$

To further analyze the sensitivity of $\gamma$, we report the results for ID (CIFAR-100) vs OOD (SVHN), ID (CIFAR-100) vs OOD (MNIST), ID (CIFAR-100) vs OOD (Texture), and ID (CIFAR-100) vs OOD (Places365) under different $\gamma$ values (0, 0.02, 0.05, 0.1). The experimental results are shown in Table 20 and Table 21. The experimental results consistently demonstrate a trend where FPR95 initially decreases and then increases as $\gamma$ rises.

A.4.11 MORE RESULTS ABOUT ID ACCURACY

We verified the ID classification performance of MSP, MSP+EMA, and their enhanced versions with our method. The experimental results in Table 22 demonstrate that our approach achieves superior ID classification performance.

A.4.12 MORE RESULTS ABOUT CALIBRATION

We evaluated the calibration performance of MSP, MSP+EMA, and their enhanced versions with our method. The experimental results in Table 22 indicate that our approach achieves superior calibration performance.

A.4.13 MORE RESULTS ABOUT ROBUSTNESS

In terms of robustness analysis, we validated the out-of-distribution (OOD) detection performance of the methods under covariate shift scenarios. The results in Table 23 demonstrate that both MSP and MSP+EMA, when enhanced with our method, achieve more robust detection performance.

Table 19: Sensitivity analysis of $\delta_2$.

| Method | CIFAR-10 | | | |
| | Near-OOD | | Far-OOD | |
| | FPR95↓ | AUROC↑ | FPR95↓ | AUROC↑ |
|---|---|---|---|---|
| $\delta_2$=0.001 | 33.10 ± 1.35 | 89.69 ± 0.49 | 20.10 ± 0.66 | 93.54 ± 0.40 |
| $\delta_2$=0.0005 | 33.69 ± 0.65 | 89.65 ± 0.24 | 21.62 ± 0.60 | 93.06 ± 0.24 |
| $\delta_2$=0.0001 | 32.47 ± 0.60 | 89.80 ± 0.28 | 21.10 ± 0.13 | 93.19 ± 0.02 |
| $\delta_2$=0.005 | 33.88 ± 0.93 | 89.60 ± 0.02 | 21.45 ± 0.69 | 92.80 ± 0.39 |
| $\delta_2$=0.01 | 36.81 ± 0.39 | 89.41 ± 0.20 | 21.45 ± 0.24 | 92.95 ± 0.18 |

Table 20: Sensitivity analysis (FPR95↓) of $\gamma$ on SVHN and MNIST.

| Method | CIFAR100 vs SVHN | | | | CIFAR100 vs MNIST | | | |
| | $\gamma = 0$ | $\gamma = 0.02$ | $\gamma = 0.05$ | $\gamma = 0.1$ | $\gamma = 0$ | $\gamma = 0.02$ | $\gamma = 0.05$ | $\gamma = 0.1$ |
|---|---|---|---|---|---|---|---|---|
| MSP+UARB | 52.46 ± 7.44 | 50.32 ± 7.88 | 50.11 ± 3.44 | 56.93 ± 1.38 | 53.70 ± 3.89 | 54.85 ± 1.31 | 55.85 ± 1.06 | 54.14 ± 1.46 |
| Energy+UARB | 45.14 ± 8.52 | 43.20 ± 9.54 | 44.07 ± 3.30 | 52.72 ± 3.23 | 52.37 ± 2.79 | 51.65 ± 1.62 | 53.27 ± 0.91 | 50.20 ± 3.68 |
| KNN+UARB | 47.18 ± 5.76 | 46.32 ± 7.30 | 43.73 ± 4.77 | 53.66 ± 7.49 | 42.36 ± 3.11 | 46.46 ± 3.99 | 46.50 ± 2.47 | 48.21 ± 7.26 |
| ReAct+UARB | 42.03 ± 6.60 | 39.90 ± 8.02 | 41.47 ± 3.33 | 50.85 ± 5.38 | 56.68 ± 2.88 | 53.34 ± 1.78 | 55.98 ± 0.43 | 51.48 ± 1.48 |
| Relation+UARB | 49.48 ± 7.64 | 47.37 ± 8.66 | 46.07 ± 5.32 | 55.24 ± 5.58 | 46.30 ± 3.53 | 48.41 ± 3.71 | 48.94 ± 1.67 | 50.33 ± 6.70 |
| Fdbd+UARB | 48.83 ± 7.57 | 46.78 ± 7.62 | 44.72 ± 4.66 | 53.87 ± 4.67 | 47.40 ± 3.41 | 49.37 ± 3.07 | 49.70 ± 2.26 | 50.31 ± 4.46 |
| Nci+UARB | 45.55 ± 5.40 | 43.53 ± 5.67 | 41.90 ± 4.61 | 48.91 ± 2.88 | 46.79 ± 2.69 | 49.71 ± 2.39 | 49.55 ± 2.70 | 49.39 ± 5.70 |

### A.4.14 More Results about Zero-order, First-order, and Second-order Difference

The primary reason for selecting the second-order difference lies in its ability to capture the acceleration of uncertainty convergence rather than just the velocity of change. First-order difference measures the instantaneous rate of change in uncertainty between consecutive epochs. While useful, it is highly sensitive to noise and short-term fluctuations in training dynamics. Second-order difference measures the change in the rate of uncertainty convergence, effectively capturing whether the model is stabilizing in its learning progress for a given instance. This makes it more robust for identifying truly "mastered" instances.

We actually did investigate the first-order difference approach, and the results are presented in Table 24. The second-order difference consistently outperforms the first-order difference across both near-OOD and far-OOD scenarios when used independently. The combination of zero-order uncertainty with second-order difference achieves the best far-OOD performance (20.10% FPR95). While the first-order difference shows competitive performance in near-OOD detection, its far-OOD performance is inferior to the second-order approach.

### A.4.15 Further Analysis on UARB with EMA

To further validate the performance of our method combined with Exponential Moving Average (EMA), we report the results of UARB+EMA integrated with different baselines at various epochs (100 and 150). The experimental results are presented in Table 25. The experimental results demonstrate that UARB+EMA achieves performance improvements with increasing training epochs, consistently across both Far-OOD and Near-OOD scenarios, and regardless of the baseline method it is combined with.

Table 21: Sensitivity analysis (FPR95↓) of $\gamma$ on Texture and Places365.

| Method | CIFAR100 vs Texture | | | | CIFAR100 vs Places365 | | | |
|---|---|---|---|---|---|---|---|---|
| | $\gamma = 0$ | $\gamma = 0.02$ | $\gamma = 0.05$ | $\gamma = 0.1$ | $\gamma = 0$ | $\gamma = 0.02$ | $\gamma = 0.05$ | $\gamma = 0.1$ |
| MSP+UARB | 62.21 ± 1.15 | 60.22 ± 0.45 | 59.97 ± 1.50 | 58.88 ± 0.29 | 56.06 ± 0.21 | 55.82 ± 0.41 | 55.99 ± 0.19 | 56.87 ± 0.88 |
| Energy+UARB | 61.31 ± 1.62 | 60.32 ± 0.69 | 61.05 ± 0.59 | 59.53 ± 0.64 | 56.87 ± 0.65 | 57.02 ± 0.67 | 57.11 ± 0.57 | 57.18 ± 0.95 |
| KNN+UARB | 57.27 ± 1.37 | 53.40 ± 2.27 | 53.17 ± 3.07 | 51.97 ± 1.67 | 60.49 ± 1.04 | 60.02 ± 0.87 | 61.14 ± 1.33 | 61.57 ± 1.26 |
| ReAct+UARB | 54.44 ± 4.29 | 52.87 ± 0.58 | 54.89 ± 0.70 | 53.85 ± 0.61 | 55.83 ± 0.51 | 55.26 ± 0.93 | 55.54 ± 0.86 | 56.25 ± 0.80 |
| Relation+UARB | 58.69 ± 2.46 | 54.55 ± 1.69 | 53.74 ± 1.85 | 53.24 ± 1.73 | 61.60 ± 1.29 | 60.77 ± 1.19 | 61.95 ± 1.45 | 62.97 ± 1.48 |
| Fdbd+UARB | 57.14 ± 2.90 | 52.84 ± 1.77 | 52.39 ± 0.28 | 53.17 ± 1.33 | 56.46 ± 0.07 | 56.26 ± 0.17 | 57.13 ± 0.63 | 57.76 ± 1.15 |
| Nci+UARB | 49.70 ± 1.50 | 47.29 ± 1.03 | 46.57 ± 0.37 | 46.08 ± 1.07 | 53.96 ± 0.34 | 53.17 ± 0.21 | 54.10 ± 0.45 | 54.39 ± 0.64 |

Table 22: ID Accuracy analysis and calibration analysis on CIFAR-10.

| Method | ID Accuracy (↑) | Calibration ECE (↓) |
|---|---|---|
| MSP | 94.96 ± 0.18 | 2.68 ± 0.12 |
| MSP+ours | 95.03 ± 0.11 | 2.62 ± 0.08 |
| MSP+EMA | 93.80 ± 0.34 | 2.69 ± 0.12 |
| MSP+EMA+ours | 94.50 ± 0.26 | 2.37 ± 0.10 |

Table 23: Robustness analysis of UARB on CIFAR-10.

| Method | FPR95 (↓) | AUROC (↑) |
|---|---|---|
| MSP | 53.29 ± 1.64 | 78.98 ± 0.83 |
| MSP+ours | 50.37 ± 1.65 | 80.97 ± 0.70 |
| MSP+EMA | 45.82 ± 0.29 | 81.37 ± 0.14 |
| MSP+EMA+ours | 44.48 ± 0.71 | 82.12 ± 0.68 |

Table 24: Analysis of zero-order, first-order, and second-order difference.

| Method | CIFAR-10 | | | |
|---|---|---|---|---|
| | Near-OOD | | Far-OOD | |
| | FPR95↓ | AUROC↑ | FPR95↓ | AUROC↑ |
| base (MSP) | 48.47 ± 0.69 | 31.90 ± 1.51 | 88.21 ± 0.16 | 90.74 ± 0.41 |
| zero-order | 37.24 ± 0.46 | 22.06 ± 0.48 | 89.32 ± 0.20 | 92.80 ± 0.04 |
| first-order | 34.47 ± 1.21 | 21.63 ± 0.70 | 89.47 ± 0.16 | 93.06 ± 0.22 |
| second-order | 32.27 ± 0.80 | 20.83 ± 1.19 | 89.89 ± 0.20 | 93.10 ± 0.33 |
| zero-order+second-order | 33.10 ± 1.35 | 20.10 ± 0.66 | 89.69 ± 0.49 | 93.54 ± 0.40 |
| zero-order+first-order | 32.41 ± 1.37 | 21.04 ± 0.99 | 89.86 ± 0.41 | 93.13 ± 0.40 |

Table 25: Comparison of OOD detection methods (with our proposed UARB) on large-scale ImageNet benchmark. All values are percentages. EMA deontes Exponential Moving Average.

| Method | Epoch | Near-OOD | | Far-OOD | |
|---|---|---|---|---|---|
| | | FPR95↓ | AUROC↑ | FPR95↓ | AUROC↑ |
| Energy+UARB+EMA | 100 | 36.90 ± 1.38 | 90.68 ± 0.24 | 19.47 ± 0.84 | 95.30 ± 0.21 |
| Energy+UARB+EMA | 150 | 32.08 ± 0.59 | 91.64 ± 0.28 | 16.57 ± 0.89 | 96.04 ± 0.10 |
| ReAct+UARB+EMA | 100 | 36.78 ± 0.66 | 90.68 ± 0.15 | 18.69 ± 1.05 | 95.44 ± 0.23 |
| ReAct+UARB+EMA | 150 | 33.92 ± 3.63 | 91.28 ± 0.81 | 15.95 ± 0.76 | 96.25 ± 0.10 |

