# OpenReview forum: "Strengthen Out-of-Distribution Detection with Uncertainty-Driven Adaptively Rectified Backpropagation"
_ICLR.cc/2026/Conference — Submitted to ICLR 2026_

### Official Review · Reviewer_EbRE · 2025-10-19

**Soundness:** 3
**Presentation:** 2
**Contribution:** 3
**Rating:** 8
**Confidence:** 4

**Summary:**

This study aims to mitigate overfitting and the consequent decline in OOD detection performance by dynamically excluding samples that the model has already mastered during training. To this end, the authors propose a novel uncertainty-based adaptive data selection strategy UARB, which leverages both zeroth- and second-order uncertainty measures and introduces class-level thresholds as data selection criteria. Comprehensive experiments conducted on both small- and large-scale datasets include evaluations in both far-OOD and near-OOD scenarios, as well as compatibility tests with a wide range of existing methods, consistently demonstrating promising performance.

**Strengths:**

**Originality:**
Unlike prior methods such as UM, which mitigate overfitting to improve OOD detection performance by performing gradient ascent on the loss over an entire batch of samples, UARB addresses overfitting at a finer, more granular level. Additionally, UARB overcomes the limitation of UM’s reliance on pretrained models, enabling direct end-to-end training.

**Quality:**
The paper is generally a good paper with a clear central idea.

**Clarity:**
The organization of the paper is good and it is easy to follow the topic and the proposed algorithms.

**Performance:**
Extensive and comprehensive experiments validate the effectiveness of UARB, demonstrating promising improvements in OOD detection performance from a data-centric perspective.

**Weaknesses:**

1. In the introduction, it would be helpful to clarify why overfitting to the training data leads to a decline in OOD detection performance. The current manuscript may be somewhat difficult to follow for readers who are not familiar with the related work.

2. Although the overall presentation is clear, several minor issues need to be addressed. For example, the purpose of the green dashed line in Figure 1 is somewhat unclear; an extra parenthesis appears in line 354; since OOD detection has already been abbreviated earlier, it may not be necessary to repeat the full term to avoid redundancy; and the captions of Tables 7 and 8 could be revised to more accurately reflect the experimental content.

3. It has not been analyzed whether the decline in OOD detection performance in the later stages of training is fully mitigated after applying the UARB training strategy.

**Questions:**

Could you further analyze whether the hyperparameter settings are sensitive to the dataset, or if they are related to the learning difficulty of the dataset?

---

> ### Author Response · Authors · 2025-11-24
> **Response to Reviewer EbRE**
>
> >W1. In the introduction, it would be helpful to clarify why overfitting to the training data leads to a decline in OOD detection performance. The current manuscript may be somewhat difficult to follow for readers who are not familiar with the related work.
>
> **Response:**
> Thank you for this insightful comment. The core issue is that an overfitted model loses its ability to generalize. It essentially "memorizes" the training data, including its atypical and specific patterns, rather than learning the general underlying features that distinguish the in-distribution classes. For OOD detection, which is a generalization task, we rely on the model producing significantly different output behaviors (e.g., lower confidence scores) for OOD samples. An overfitted model may produce high confidence even for inputs that are far from the training data, leading to failures in identifying them as OOD. In the revised manuscript, we will  incorporate this explanation to improve the flow and clarity of the introduction.
>
> > W2. Although the overall presentation is clear, several minor issues need to be addressed. For example, the purpose of the green dashed line in Figure 1 is somewhat unclear; an extra parenthesis appears in line 354; since OOD detection has already been abbreviated earlier, it may not be necessary to repeat the full term to avoid redundancy; and the captions of Tables 7 and 8 could be revised to more accurately reflect the experimental content.
>
> **Response:**  We have carefully addressed all the points you raised, as detailed below:
> 1. We have identified a minor inaccuracy in the original green dashed line of Figure 1 and have subsequently corrected it. The line is intended to represent the point in time at which the In-Distribution (ID) accuracy reaches its peak.
> 2. We have removed the extra parenthesis in Line 354.
> 3. We have revised the manuscript and used the abbreviation "OOD" consistently after its first full definition.
> 4. We have revised the captions of Tables 7 and 8 to guide the reader to interpret the tables' content and significance.
>
> > W3. It has not been analyzed whether the decline in OOD detection performance in the later stages of training is fully mitigated after applying the UARB training strategy.
>
> **Response:**
> As suggested, we provide a detailed analysis and empirical evidence to address this point. To directly investigate whether UARB mitigates the performance decline in later training stages, we conducted a systematic evaluation comparing the OOD detection performance of baseline methods versus their UARB-enhanced versions across different training phases. Fig 2 in the revised manuscript demonstrates the average FPR95 of the baseline and the UARB-enhanced method under the Far-OOD scenario at different later stages of training. The results reveals that UARB significantly alleviates the degradation in OOD detection performance during later training stages.
>
> > Q1. Could you further analyze whether the hyperparameter settings are sensitive to the dataset, or if they are related to the learning difficulty of the dataset?
>
> **Response:**
> Thank you for this insightful question. Since different datasets vary in terms of the number of classes, sample size, and image resolution, we have correspondingly adjusted hyperparameters for each dataset to ensure optimal adaptation to their specific characteristics. The most important parameter in our UARB $\gamma$, is used to balance the adaptive term, and its value varies across different datasets. Specifically, it is set to 0.1 for CIFAR-10, 0.02 for CIFAR-100, and 0.1 for ImageNet.

---

> > ### Comment · Reviewer_EbRE · 2025-11-24
> >
> > Thank you for your reply. For now, I’ve decided to retain my positive evaluation.

---

> > > ### Author Response · Authors · 2025-11-24
> > > **Thank you for recognizing our strengths.**
> > >
> > > Dear Reviewer EbRE,
> > >
> > > Thank you for your reply. We appreciate your recognition on the originality, quality, performance, and clarity of our work.
> > >
> > > Best,
> > > Authors

---

### Official Review · Reviewer_LZKj · 2025-10-31

**Soundness:** 2
**Presentation:** 2
**Contribution:** 2
**Rating:** 2
**Confidence:** 4

**Summary:**

This paper addresses the performance degradation in the late training stage due to overfitting on mastered in-distribution samples. The authors propose a data-centric end-to-end framework UARB that identifies mastered samples and then implements instance-level early stopping by excluding mastered samples from backpropagation. Experiments on CIFAR-10/100 and ImageNet show UARB improves OOD detection when combined with baselines like MSP, Energy, and KNN.

**Strengths:**

- Unlike model-centric methods that treat symptoms of overfitting, UARB dynamically filters training data, which aligning with the growing focus on data quality in OOD detection.
- By integrating class-wise uncertainty variance, UARB balances training of easy and hard classes.

**Weaknesses:**

- Most experimental improvements are margin, as can be observed in the tables, especially on ImageNet dataset. Such gains are insufficient to justify the added complexity over simpler baselines.
- The paper claims UARB mitigates overfitting but there is no direct visualization of overfitting, which is critical to validate its core mechanism.
- Experiments only use ResNet-18, a lightweight but outdated architecture. Modern OOD detection increasingly relies on transformers (e.g., ViT, Swin Transformer) or pre-trained models. UARB’s effectiveness on these architectures is unproven, limiting its practical relevance.
- UARB adds non-trivial computations, e.g., second-order difference across epochs, class-wise variance calculation, but provides no analysis of computational cost compared to baselines. For resource-constrained edge devices, this omission makes it impossible to assess practicality.
- The paper relies entirely on experimental results. There is no theoretical guarantees for the proposed method.

**Questions:**

None

---

> ### Author Response · Authors · 2025-11-24
> **Response to Reviewer LZKj (part 1)**
>
> > W1. Most experimental improvements are margin, as can be observed in the tables, especially on ImageNet dataset. Such gains are insufficient to justify the added complexity over simpler baselines.
>
> **Response:** We respectfully disagree and argue that our improvements are consistent and significant over existing methods, for the following reasons:
> 1. UARB improves performance for nearly all baseline methods on all three datasets in both near-OOD and far-OOD scenarios. This consistency demonstrates the robustness and general applicability of our approach.
> 2. The performance gains achieved by UARB are particularly notable when considered in the context of recent literature. For example, in the recently published work [1] which addresses a similar problem, the reported improvements on ImageNet are smaller than CIFAR, yet are considered significant contributions to the field. This suggests that in the challenging ImageNet benchmark, even modest improvements represent meaningful advancements.
> [1] Yang, Y., & Xu, H. Strengthen Out-of-Distribution Detection Capability with Progressive Self-Knowledge Distillation. In Forty-second International Conference on Machine Learning.
>
>
>
> > W2. The paper claims UARB mitigates overfitting but there is no direct visualization of overfitting, which is critical to validate its core mechanism.
>
> **Response:**
> To directly investigate whether UARB mitigates the overfitting in later training stages, we conducted a systematic evaluation comparing the OOD detection performance of baseline methods versus their UARB-enhanced versions across different training phases. Fig 2 in the revised manuscript demonstrates the average FPR95 of the baseline and the UARB-enhanced version at different later stages of training. The results reveals that UARB significantly alleviates the overfitting during later training stages.
>
> Furthermore, we visualized the t-SNE plots comparing the baseline method (Fig 3(a) in the revised manuscript) and our enhanced method (Fig 3 (b) in the revised manuscript). The results clearly demonstrate that our method achieves superior separation between in-distribution (ID) and out-of-distribution (OOD) data. This observation provides visual confirmation that overfitting has been effectively mitigated to a certain extent.

---

> ### Author Response · Authors · 2025-11-24
> **Response to Reviewer LZKj (part 2)**
>
> > W3. Experiments only use ResNet-18, a lightweight but outdated architecture. Modern OOD detection increasingly relies on transformers (e.g., ViT, Swin Transformer) or pre-trained models. UARB’s effectiveness on these architectures is unproven, limiting its practical relevance.
>
> **Response:**
> We did not employ Transformer models such as ViT because their performance on CIFAR is poor due to the low-resolution images. To further investigate the effectiveness of UARB, we also conducted validation on ResNet-50. The experimental results below demonstrate that incorporating UARB leads to comprehensive performance improvements in almost baseline methods.
>
> | Method       | CIFAR-100 | TIN   | Near-Avg | MNIST  | SVHN   | Texture | Places365 | Far-Avg |
> |-------------|----------|-------|---------|--------|--------|---------|-----------|--------|
> | MSP         | 51.35 ± 1.58 | 47.70 ± 1.65 | 49.52 ± 1.61 | 42.02 ± 4.93 | 44.27 ± 5.56 | 48.39 ± 1.82 | 46.54 ± 1.53 | 45.30 ± 1.91 |
> | MSP+ours    | 47.14 ± 0.15 | 43.19 ± 0.04 | 45.17 ± 0.05 | 39.10 ± 4.89 | 35.09 ± 4.16 | 43.47 ± 0.51 | 43.21 ± 0.30 | 40.22 ± 1.83 |
> | Energy         | 54.42 ± 2.60 | 48.58 ± 2.78 | 51.50 ± 2.68 | 36.52 ± 5.20 | 52.06 ± 8.05 | 49.25 ± 3.56 | 44.47 ± 3.50 | 45.58 ± 4.48 |
> | Energy+ours    | 47.16 ± 0.46 | 41.31 ± 0.45 | 44.24 ± 0.43 | 29.10 ± 6.57 | 39.81 ± 4.30 | 44.09 ± 0.54 | 38.31 ± 0.80 | 37.83 ± 2.26 |
> | KNN         | 51.04 ± 1.41 | 45.59 ± 0.51 | 48.31 ± 0.87 | 33.75 ± 6.81 | 45.65 ± 2.99 | 44.39 ± 1.10 | 43.47 ± 1.32 | 41.81 ± 1.35 |
> | KNN+ours    | 48.99 ± 0.85 | 43.11 ± 1.02 | 46.05 ± 0.93 | 31.77 ± 3.30 | 38.54 ± 2.02 | 43.40 ± 1.90 | 42.36 ± 0.29 | 39.02 ± 0.64 |
> | ReAct       | 66.70 ± 0.56 | 64.43 ± 1.22 | 65.57 ± 0.68 | 68.85 ± 3.06 | 69.58 ± 7.90 | 66.94 ± 3.06 | 56.39 ± 0.47 | 65.44 ± 2.36 |
> | ReAct+ours  | 61.04 ± 2.12 | 57.70 ± 2.72 | 59.37 ± 2.42 | 56.82 ± 8.21 | 59.21 ± 5.76 | 62.12 ± 2.10 | 50.73 ± 3.18 | 57.22 ± 2.93 |
> | Relation    | 55.91 ± 1.11 | 51.27 ± 2.24 | 53.59 ± 1.62 | 41.28 ± 8.47 | 41.97 ± 3.03 | 48.30 ± 2.86 | 52.61 ± 2.41 | 46.04 ± 0.75 |
> | Relation+ours | 57.33 ± 3.02 | 50.91 ± 4.30 | 54.12 ± 3.63 | 35.03 ± 2.05 | 36.13 ± 2.87 | 56.95 ± 9.57 | 54.84 ± 1.41 | 45.74 ± 3.51 |
> | Fdbd        | 59.47 ± 0.97 | 51.95 ± 0.54 | 55.71 ± 0.73 | 39.62 ± 11.92 | 62.51 ± 4.82 | 52.15 ± 3.36 | 47.49 ± 2.15 | 50.44 ± 2.56 |
> | Fdbd+ours   | 51.51 ± 1.14 | 45.16 ± 1.30 | 48.33 ± 1.21 | 34.84 ± 4.46 | 53.74 ± 5.92 | 45.03 ± 2.06 | 40.19 ± 0.75 | 43.45 ± 2.19 |
> | Nci         | 59.41 ± 2.39 | 55.85 ± 2.06 | 57.63 ± 2.23 | 44.96 ± 6.21 | 43.07 ± 6.11 | 58.55 ± 1.92 | 56.54 ± 3.81 | 50.78 ± 1.31 |
> | Nci+ours    | 59.81 ± 0.99 | 54.38 ± 1.01 | 57.10 ± 0.75 | 45.35 ± 12.11 | 40.91 ± 4.31 | 60.83 ± 5.05 | 56.98 ± 1.35 | 51.02 ± 3.70 |
> | DICE        | 56.24 ± 3.50 | 49.75 ± 3.92 | 53.00 ± 3.71 | 35.14 ± 10.68 | 48.63 ± 5.52 | 50.28 ± 5.67 | 48.32 ± 5.17 | 45.59 ± 5.82 |
> | DICE+ours   | 48.77 ± 0.55 | 42.52 ± 0.64 | 45.64 ± 0.57 | 25.03 ± 6.57 | 36.04 ± 6.69 | 46.29 ± 1.94 | 41.40 ± 1.35 | 37.19 ± 2.70 |
> | Scale       | 55.55 ± 3.95 | 50.65 ± 4.17 | 53.10 ± 4.06 | 45.79 ± 3.73 | 50.97 ± 9.42 | 52.08 ± 6.62 | 47.29 ± 4.75 | 49.03 ± 5.98 |
> | Scale+ours  | 48.37 ± 0.32 | 42.43 ± 0.41 | 45.40 ± 0.36 | 33.75 ± 6.51 | 39.96 ± 5.20 | 45.68 ± 2.54 | 40.11 ± 1.62 | 39.87 ± 2.18 |
>
>
> > W4. UARB adds non-trivial computations, e.g., second-order difference across epochs, class-wise variance calculation, but provides no analysis of computational cost compared to baselines. For resource-constrained edge devices, this omission makes it impossible to assess practicality.
>
> **Response:**
> Thank you for raising this important concern regarding the computational and memory overhead of UARB. We rigorously measured the training time and GPU memory usage for a standard benchmark (CIFAR-10 with a ResNet-18 backbone) to quantify the overhead. The results are as follows: (1) Baseline Method:​ 2030 seconds, 1452 MiB GPU memory; (2) Our Method (UARB):​ 2371 seconds, 1456 MiB GPU memory. The additional 341 seconds of training time in UARB is a relatively small cost in the context of a full training run. We adhere to the first-in-first-out principle and utilize a queue to track the uncertainty of each sample, which does not significantly increase memory costs.
>
> > W5. The paper relies entirely on experimental results. There is no theoretical guarantees for the proposed method.
>
> **Response:**
> We acknowledge that our paper primarily relies on empirical validation, and we appreciate your emphasis on the importance of theoretical analysis. Our approach is consistent with several influential works in OOD detection, which also prioritize empirical validation over theoretical guarantees. Moreover, the OOD detection task itself presents unique challenges for theoretical analysis. While theoretical guarantees are desirable, we believe that comprehensive empirical validation is a robust alternative for evaluating OOD detection methods.

---

### Official Review · Reviewer_PAyv · 2025-11-01

**Soundness:** 3
**Presentation:** 2
**Contribution:** 2
**Rating:** 4
**Confidence:** 3

**Summary:**

Overfitting during the training phase can significantly degrade OOD detection performance. This paper addresses this issue by proposing Uncertainty-driven Adaptively Rectified Backpropagation (UARB), which mitigates overfitting by blocking the backpropagation of mastered instances. The authors first demonstrate that OOD detection performance declines in the later stages of training due to overfitting. To alleviate this, they propose UARB, which leverages uncertainty to estimate the degree of learning for each instance and selectively stops training for sufficiently learned samples. Furthermore, they highlight the issue of class imbalance during training and introduce an adaptive threshold mechanism, which employs both the uncertainty and its second-order difference to dynamically balance per-class thresholds when identifying mastered instances. Extensive experiments across various datasets and OOD detection scenarios demonstrate the effectiveness and generality of their method compared with UM.

**Strengths:**

1. UARB is a plug-and-play method with strong applicability to various existing OOD detection approaches.
2. The problem of overfitting is clearly demonstrated through experimental evidence (as shown in Figure 1).
3. The motivation behind using uncertainty, the second-order difference, and the adaptive threshold mechanism is clearly explained.
4. UARB clearly outperforms UM, the most relevant prior approach, across multiple benchmarks.

**Weaknesses:**

1. In Section 2.2, the authors mention that they empirically verify the increase in instances that meet the mastered criteria. However, no corresponding experimental results are presented in the paper.
2. The authors argue that using a fixed threshold to determine when to exclude an instance from backpropagation is theoretically unsound, yet no further explanation or theoretical analysis is provided. I encourage the authors to elaborate on this point with more detailed justification.
3. UARB requires tracking the uncertainty across three consecutive epochs for the entire training dataset, which introduces substantial computational and memory overhead — especially as the dataset size and model complexity increase.
4. Although this paper mainly focuses on OOD detection during the training phase, it omits discussion of more recent and powerful OOD detection methods, such as DICE [1] and LINe [2].
5. While variance normalization can provide certain benefits, it assumes that all classes are generally imbalanced. In early training stages or fine-grained datasets where class-wise variation is minimal, this assumption may instead lead to adverse effects.
6. I recommend that the authors revise the full manuscript to improve readability and completeness.
- In the Introduction, some expressions (e.g., mastered instance, data-level issue) are difficult to understand immediately.
- Minor typos:
 Line 269: scenatios → scenarios
 Line 411: Table → Equation
- In the main manuscript, Table 7 is not referenced or used.

[1] "Dice: Leveraging sparsification for out-of-distribution detection." ECCV 2022.\
[2] "LINe: Out-of-distribution detection by leveraging important neurons." CVPR 2023.

**Questions:**

1. Extending from the concern about computational cost, I am curious about the applicability of the proposed method to larger backbones, such as ResNet-50 or ViT.
2. UARB utilizes both uncertainty and the second-order difference. Could the authors clarify why the first-order difference was not considered in their formulation? (or provide some experimental results).
3. Since UARB uses uncertainty from three consecutive epochs to compute the second-order difference, I recommend conducting an ablation study on this hyperparameter to analyze its sensitivity and impact.

---

> ### Author Response · Authors · 2025-11-24
> **Response to Reviewer PAyv (part 1)**
>
> > W1. In Section 2.2, the authors mention that they empirically verify the increase in instances that meet the mastered criteria. However, no corresponding experimental results are presented in the paper.
>
> **Response:**  As suggested, we have added a figure that explicitly demonstrates this empirical verification. Fig 4 in the revisied manuscript illustrates the number of mastered instances throughout the training process on the CIFAR-10 dataset. Through it, we can observe that the number of mastered instances increases slowly at first, then rises rapidly, and finally plateaus.
>
> > W2. The authors argue that using a fixed threshold to determine when to exclude an instance from backpropagation is theoretically unsound, yet no further explanation or theoretical analysis is provided. I encourage the authors to elaborate on this point with more detailed justification.
>
> **Response:**
> Thank you for this valuable feedback regarding the justification for our adaptive threshold approach.
>
> As shown in Table 1, explicitly modeling class-specific characteristics through variance consistency constraints leads to consistent improvements in OOD detection performance. This empirical evidence strongly suggests that accounting for inter-class differences is essential. A fixed threshold inherently ignores these critical variations, leading to suboptimal model behavior.
>
> Fixed thresholds cause uneven learning across categories. For difficult classes with high uncertainty variance, a fixed threshold may be too strict, prematurely excluding instances before they are truly mastered. Conversely, for easy classes with low uncertainty variance, the same threshold may be too lenient, retaining instances long after mastery and potentially causing overfitting. This imbalance disrupts optimal learning across all classes.
>
> The experimental results in Table 4 provide direct comparative evidence: our adaptive threshold method (UARB) consistently outperforms the fixed threshold approach (URB) across multiple OOD detection benchmarks. This performance gap empirically validates the advantage of adaptive criteria to each class's actual learning state, rather than applying a rigid, universal standard.
>
> > W3. UARB requires tracking the uncertainty across three consecutive epochs for the entire training dataset, which introduces substantial computational and memory overhead — especially as the dataset size and model complexity increase.
>
> **Response:**
> Thank you for raising this important concern regarding the computational and memory overhead of UARB. We rigorously measured the training time and GPU memory usage for a standard benchmark (CIFAR-10 with a ResNet-18 backbone) to quantify the overhead. The results are as follows: (1) Baseline Method:​ 2030 seconds, 1452 MiB GPU memory; (2) Our Method (UARB):​ 2371 seconds, 1456 MiB GPU memory. The additional 341 seconds of training time in UARB is a relatively small cost in the context of a full training run. We adhere to the first-in-first-out principle and utilize a queue to track the uncertainty of each sample, which does not significantly increase memory costs.

---

> ### Author Response · Authors · 2025-11-24
> **Response to Reviewer PAyv (part 2)**
>
> >W4. Although this paper mainly focuses on OOD detection during the training phase, it omits discussion of more recent and powerful OOD detection methods, such as DICE and LINe.
>
> **Response:**
> Based on your suggestions, we have newly added two baselines. Since LINe is not available in the OpenOOD library, we replaced LINe with Scale [1]. The experimental results are as follows. Experimental results demonstrate that both DICE and Scale exhibit significant improvements in FPR95 and AUROC under both near-OOD and far-OOD scenarios after incorporating UARB.
>
> [1]Xu, Kai, et al. "Scaling for Training Time and Post-hoc Out-of-distribution Detection Enhancement." The Twelfth International Conference on Learning Representations.
>
> | Method       | CIFAR-100 | TIN   | Near-Avg | MNIST  | SVHN   | Texture | Places365 | Far-Avg |
> |--------------|----------|-------|---------|--------|--------|---------|-----------|--------|
> | **FPR95**    |          |       |         |        |        |         |           |        |
> | DICE         | 76.17 ± 3.28 | 68.56 ± 5.06 | 72.36 ± 4.02 | 21.36 ± 2.55 | 32.11 ± 2.68 | 56.99 ± 4.53 | 81.73 ± 8.72 | 48.05 ± 2.55 |
> | DICE+ours    | 66.20 ± 2.22 | 58.23 ± 4.46 | 62.21 ± 3.14 | 7.61 ± 1.94 | 9.46 ± 4.37 | 51.78 ± 4.62 | 67.11 ± 13.08 | 33.99 ± 0.96 |
> | Scale        | 83.86 ± 1.45 | 80.96 ± 1.28 | 82.41 ± 1.37 | 51.44 ± 9.24 | 72.30 ± 12.01 | 86.84 ± 4.39 | 70.61 ± 3.43 | 70.30 ± 5.92 |
> | Scale+ours   | 62.91 ± 3.33 | 53.93 ± 3.20 | 58.42 ± 3.24 | 18.11 ± 2.82 | 12.40 ± 1.56 | 48.29 ± 11.90 | 42.47 ± 3.67 | 30.32 ± 2.95 |
> | **AUROC**    |          |       |         |        |        |         |           |        |
> | DICE         | 79.17 ± 0.70 | 81.65 ± 1.79 | 80.41 ± 1.19 | 94.43 ± 2.00 | 90.68 ± 1.38 | 84.43 ± 0.53 | 75.88 ± 4.49 | 86.35 ± 1.52 |
> | DICE+ours    | 81.53 ± 0.67 | 83.19 ± 1.15 | 82.36 ± 0.84 | 98.24 ± 0.51 | 97.27 ± 1.43 | 84.87 ± 1.96 | 80.94 ± 4.13 | 90.33 ± 0.12 |
> | Scale        | 79.92 ± 0.61 | 82.94 ± 0.42 | 81.43 ± 0.52 | 90.62 ± 1.60 | 82.85 ± 3.83 | 81.16 ± 1.72 | 86.28 ± 1.59 | 85.23 ± 1.69 |
> | Scale+ours   | 86.11 ± 0.97 | 88.78 ± 0.66 | 87.45 ± 0.80 | 95.22 ± 0.59 | 96.66 ± 0.79 | 90.19 ± 1.95 | 90.99 ± 0.89 | 93.26 ± 0.47 |
> |
>
> > W5. While variance normalization can provide certain benefits, it assumes that all classes are generally imbalanced. In early training stages or fine-grained datasets where class-wise variation is minimal, this assumption may instead lead to adverse effects.
>
> **Response:**  We clarify that our approach does not heavily assume general class imbalance. Instead, the Variance Consistency (VC) constraint is designed to address the inherent variation in learning dynamics across classes, which exists even in class-balanced scenarios. This is empirically validated in Table 1​ of our paper, where we tested the VC constraint on a class-balanced dataset. The results demonstrate consistent improvements in OOD detection performance (e.g., reduction in FPR95 and increase in AUROC) for both near-OOD and far-OOD cases, indicating that class-wise variance normalization provides benefits regardless of imbalance. This suggests that the key issue is not imbalance itself, but the divergence in optimization progress and uncertainty patterns across classes.
>
> It is possible that, in early training stages or fine-grained datasets, class-wise variation might be minimal, potentially leading to adverse effects if variance normalization is applied naively. To mitigate this, we incorporate a warm-up phase​ before fully applying the VC constraint. Specifically, during early training, the model is allowed to learn without variance normalization to avoid premature and noisy estimates. This warm-up period ensures that the model reaches a stable state where class-wise variances are meaningful. Only after this phase do we activate the variance normalization.
>
> > W6. I recommend that the authors revise the full manuscript to improve readability and completeness. For example, in the Introduction, some expressions (e.g., mastered instance, data-level issue) are difficult to understand immediately. Minor typos: Line 269: scenatios → scenarios Line 411: Table → Equation. In the main manuscript, Table 7 is not referenced or used.
>
> **Response:**  As suggested, we have fixed the typos and improved the presentation.

---

> ### Author Response · Authors · 2025-11-24
> **Response to Reviewer PAyv (part 3)**
>
> > Q1. I am curious about the applicability of the proposed method to larger backbones, such as ResNet-50 or ViT.
>
> **Response:** To further assess the applicability of UARB, we conducted experiments using ResNet-50 as the backbone. The results below demonstrate that integrating UARB consistently improves OOD detection performance across nearly all baseline methods.
>
>
> | Method       | CIFAR-100 | TIN   | Near-Avg | MNIST  | SVHN   | Texture | Places365 | Far-Avg |
> |-------------|----------|-------|---------|--------|--------|---------|-----------|--------|
> | MSP         | 51.35 ± 1.58 | 47.70 ± 1.65 | 49.52 ± 1.61 | 42.02 ± 4.93 | 44.27 ± 5.56 | 48.39 ± 1.82 | 46.54 ± 1.53 | 45.30 ± 1.91 |
> | MSP+ours    | 47.14 ± 0.15 | 43.19 ± 0.04 | 45.17 ± 0.05 | 39.10 ± 4.89 | 35.09 ± 4.16 | 43.47 ± 0.51 | 43.21 ± 0.30 | 40.22 ± 1.83 |
> | Energy         | 54.42 ± 2.60 | 48.58 ± 2.78 | 51.50 ± 2.68 | 36.52 ± 5.20 | 52.06 ± 8.05 | 49.25 ± 3.56 | 44.47 ± 3.50 | 45.58 ± 4.48 |
> | Energy+ours    | 47.16 ± 0.46 | 41.31 ± 0.45 | 44.24 ± 0.43 | 29.10 ± 6.57 | 39.81 ± 4.30 | 44.09 ± 0.54 | 38.31 ± 0.80 | 37.83 ± 2.26 |
> | KNN         | 51.04 ± 1.41 | 45.59 ± 0.51 | 48.31 ± 0.87 | 33.75 ± 6.81 | 45.65 ± 2.99 | 44.39 ± 1.10 | 43.47 ± 1.32 | 41.81 ± 1.35 |
> | KNN+ours    | 48.99 ± 0.85 | 43.11 ± 1.02 | 46.05 ± 0.93 | 31.77 ± 3.30 | 38.54 ± 2.02 | 43.40 ± 1.90 | 42.36 ± 0.29 | 39.02 ± 0.64 |
> | ReAct       | 66.70 ± 0.56 | 64.43 ± 1.22 | 65.57 ± 0.68 | 68.85 ± 3.06 | 69.58 ± 7.90 | 66.94 ± 3.06 | 56.39 ± 0.47 | 65.44 ± 2.36 |
> | ReAct+ours  | 61.04 ± 2.12 | 57.70 ± 2.72 | 59.37 ± 2.42 | 56.82 ± 8.21 | 59.21 ± 5.76 | 62.12 ± 2.10 | 50.73 ± 3.18 | 57.22 ± 2.93 |
> | Relation    | 55.91 ± 1.11 | 51.27 ± 2.24 | 53.59 ± 1.62 | 41.28 ± 8.47 | 41.97 ± 3.03 | 48.30 ± 2.86 | 52.61 ± 2.41 | 46.04 ± 0.75 |
> | Relation+ours | 57.33 ± 3.02 | 50.91 ± 4.30 | 54.12 ± 3.63 | 35.03 ± 2.05 | 36.13 ± 2.87 | 56.95 ± 9.57 | 54.84 ± 1.41 | 45.74 ± 3.51 |
> | Fdbd        | 59.47 ± 0.97 | 51.95 ± 0.54 | 55.71 ± 0.73 | 39.62 ± 11.92 | 62.51 ± 4.82 | 52.15 ± 3.36 | 47.49 ± 2.15 | 50.44 ± 2.56 |
> | Fdbd+ours   | 51.51 ± 1.14 | 45.16 ± 1.30 | 48.33 ± 1.21 | 34.84 ± 4.46 | 53.74 ± 5.92 | 45.03 ± 2.06 | 40.19 ± 0.75 | 43.45 ± 2.19 |
> | Nci         | 59.41 ± 2.39 | 55.85 ± 2.06 | 57.63 ± 2.23 | 44.96 ± 6.21 | 43.07 ± 6.11 | 58.55 ± 1.92 | 56.54 ± 3.81 | 50.78 ± 1.31 |
> | Nci+ours    | 59.81 ± 0.99 | 54.38 ± 1.01 | 57.10 ± 0.75 | 45.35 ± 12.11 | 40.91 ± 4.31 | 60.83 ± 5.05 | 56.98 ± 1.35 | 51.02 ± 3.70 |
> | DICE        | 56.24 ± 3.50 | 49.75 ± 3.92 | 53.00 ± 3.71 | 35.14 ± 10.68 | 48.63 ± 5.52 | 50.28 ± 5.67 | 48.32 ± 5.17 | 45.59 ± 5.82 |
> | DICE+ours   | 48.77 ± 0.55 | 42.52 ± 0.64 | 45.64 ± 0.57 | 25.03 ± 6.57 | 36.04 ± 6.69 | 46.29 ± 1.94 | 41.40 ± 1.35 | 37.19 ± 2.70 |
> | Scale       | 55.55 ± 3.95 | 50.65 ± 4.17 | 53.10 ± 4.06 | 45.79 ± 3.73 | 50.97 ± 9.42 | 52.08 ± 6.62 | 47.29 ± 4.75 | 49.03 ± 5.98 |
> | Scale+ours  | 48.37 ± 0.32 | 42.43 ± 0.41 | 45.40 ± 0.36 | 33.75 ± 6.51 | 39.96 ± 5.20 | 45.68 ± 2.54 | 40.11 ± 1.62 | 39.87 ± 2.18 |
> |

---

> ### Author Response · Authors · 2025-11-24
> **Response to Reviewer PAyv (part 4)**
>
> >Q2. UARB utilizes both uncertainty and the second-order difference. Could the authors clarify why the first-order difference was not considered in their formulation? (or provide some experimental results).
>
> **Response:**
> The primary reason for selecting the second-order difference lies in its ability to capture the acceleration of uncertainty convergence rather than just the velocity of change. First-order difference measures the instantaneous rate of change in uncertainty between consecutive epochs. While useful, it is highly sensitive to noise and short-term fluctuations in training dynamics. Second-order difference measures the change in the rate of uncertainty convergence, effectively capturing whether the model is stabilizing in its learning progress for a given instance. This makes it more robust for identifying truly "mastered" instances.
>
> We actually did investigate the first-order difference approach, and the results are presented as follows. The second-order difference consistently outperforms the first-order difference across both near-OOD and far-OOD scenarios when used independently. The combination of zero-order uncertainty with second-order difference achieves the best far-OOD performance (20.10% FPR95). While the first-order difference shows competitive performance in near-OOD detection, its far-OOD performance is inferior to the second-order approach.
>
> | Method       | Near-Avg (FPR95) | Far-Avg (FPR95) | Near-Avg (AUROC) | Far-Avg (AUROC) |
> |--------------|-----------------|----------------|-----------------|----------------|
> | base (MSP)   | 48.47 ± 0.69    | 31.90 ± 1.51   | 88.21 ± 0.16    | 90.74 ± 0.41   |
> | zero-order            | 37.24 ± 0.46    | 22.06 ± 0.48   | 89.32 ± 0.20    | 92.80 ± 0.04   |
> | first-order            | 34.47 ± 1.21    | 21.63 ± 0.70   | 89.47 ± 0.16    | 93.06 ± 0.22   |
> | second-order            | 32.27 ± 0.80    | 20.83 ± 1.19   | 89.89 ± 0.20    | 93.10 ± 0.33   |
> | zero-order+second-order          | 33.10 ± 1.35    | 20.10 ± 0.66   | 89.69 ± 0.49    | 93.54 ± 0.40   |
> | zero-order+first-order          | 32.41 ± 1.37    | 21.04 ± 0.99   | 89.86 ± 0.41    | 93.13 ± 0.40   |
> |
>
> > Q3. Since UARB uses uncertainty from three consecutive epochs to compute the second-order difference, I recommend conducting an ablation study on this hyperparameter to analyze its sensitivity and impact.
>
>
> **Response:**
> Thank you for this excellent suggestion regarding the ablation study on the hyperparameter $\delta_2$ used in our second-order difference calculation. We have conducted a comprehensive sensitivity analysis as follows. We individually employed zero-order and second-order metrics to validate their respective impacts. As illustrated in the table below, the second-order metric demonstrates superior OOD detection performance in both near-OOD and far-OOD scenarios.
>
> | Method       | Near-Avg (FPR95) | Far-Avg (FPR95) | Near-Avg (AUROC) | Far-Avg (AUROC) |
> |--------------|-----------------|----------------|-----------------|----------------|
> | base (MSP)   | 48.47 ± 0.69    | 31.90 ± 1.51   | 88.21 ± 0.16    | 90.74 ± 0.41   |
> | zero-order            | 37.24 ± 0.46    | 22.06 ± 0.48   | 89.32 ± 0.20    | 92.80 ± 0.04   |
> | second-order            | 32.27 ± 0.80    | 20.83 ± 1.19   | 89.89 ± 0.20    | 93.10 ± 0.33   |
> | zero-order+second-order          | 33.10 ± 1.35    | 20.10 ± 0.66   | 89.69 ± 0.49    | 93.54 ± 0.40   |
> |
>
> To further verify the sensitivity of the second-order difference, we adjusted the hyperparameter $\delta_2$ that controls it. According to the experimental results, the second-order difference is not particularly sensitive to the choice of $\delta_2$.
>
> |    | Near-Avg (FPR95) | Far-Avg (FPR95) | Near-Avg (AUROC) | Far-Avg (AUROC) |
> |---------------|----------------|----------------|-----------------|----------------|
> | $\delta_2$=0.001 | 33.10 ± 1.35   | 20.10 ± 0.66   | 89.69 ± 0.49    | 93.54 ± 0.40   |
> | $\delta_2$=0.0005 | 33.69 ± 0.65   | 21.62 ± 0.60   | 89.65 ± 0.24    | 93.06 ± 0.24   |
> | $\delta_2$=0.0001 | 32.47 ± 0.60   | 21.10 ± 0.13   | 89.80 ± 0.28    | 93.19 ± 0.02   |
> | $\delta_2$=0.005 | 33.88 ± 0.93   | 21.45 ± 0.69   | 89.60 ± 0.02    | 92.80 ± 0.39   |
> | $\delta_2$=0.01 | 36.81 ± 0.39   | 21.45 ± 0.24   | 89.41 ± 0.20    | 92.95 ± 0.18   |
> |

---

### Official Review · Reviewer_LSjt · 2025-11-01

**Soundness:** 3
**Presentation:** 2
**Contribution:** 3
**Rating:** 4
**Confidence:** 3

**Summary:**

In this paper, the authors propose Uncertainty‑driven Adaptively Rectified Backpropagation (UARB) to improve OOD detection performance of classification models. The motivation is that there is an issue that during training the model may over-fit to ID instances in later epochs, thereby degrading its ability to distinguish OOD. The authors monitor the uncertainty score (zero-order) and the second-order difference of uncertainty (roughly the change in change) over epochs. If both fall below thresholds, they declare the instance mastered. Once an instance is marked mastered, they exclude it from further back-propagation so as to prevent over-training on that sample and reduce over-fitting. They use class-informed statistics to derive adaptive thresholds for mastering each class. They report experiments showing improved OOD detection across benchmarks when using UARB compared with baseline training.

**Strengths:**

- The observation that OOD detection performance can decline during later training epochs is important and less emphasized in many OOD works.
- Intervening in the training loop is a novel angle for OOD detection.
- By recognizing that different classes converge at different rates, the authors design class-specific thresholds.

**Weaknesses:**

- The methodology relies heavily on an uncertainty score per instance and its second-order difference across epochs. But the paper is somewhat light on which uncertainty metric is used. How sensitive results are to this choice, and how robustly it correlates with mastery.
- Is the second order difference stable and sufficiently noise-free to reliably identify mastered instances? How does noise in uncertainty estimates affect the criteria?
- The details of how the class-specific thresholds are computed are relatively brief. How sensitive are results to the thresholds
- By excluding mastered samples from further training, the method effectively reduces the effective training set size and focuses training on harder instances. While this may reduce over-fitting, it could potentially under-fit some classes or lead to class imbalance issues. Is there analysis of whether ID classification accuracy or calibration suffers?
- It would be great to add the following ablation studies.
1. the effect of using only zero-order vs. also second-order difference
2. fixed threshold vs adaptive threshold;
3. baseline training without UARB but with same number of epochs/training budget.

**Questions:**

- It would be great to add discussion on uncertainty metrics (entropy, MSP, and loss) and show how the criteria for mastery behaves under each.
- It would be great to include sensitivity analyses on δ₁, δ₂ and show robustness of performance
- It would be great to add experiments on more diverse datasets and OOD types including near-OOD/far-OOD
- It should report the impact on primary classification task metrics (ID accuracy, calibration, robustness)

---

> ### Author Response · Authors · 2025-11-24
> **Response to Reviewer LSjt (part1)**
>
> > W1&Q1. which uncertainty metric is used. How sensitive results are to this choice, and how robustly it correlates with mastery.  It would be great to add discussion on uncertainty metrics (entropy, MSP, and loss) and show how the criteria for mastery behaves under each.
>
> **Response:** We clarified that, in Table 8 of our paper (Section 3.3), we have provided an empirical comparison of different uncertainty metrics within our UARB framework. The core of our mastery criterion is the principle that a mastered instance should exhibit **low and stable uncertainty**. This is formalized by requiring both the uncertainty value $U_i(w^{(t)})$ and the absolute value of its second-order difference $|\Delta^2 U_i(w^{(t)})|$ to fall below small, positive thresholds ($\delta_1$ and $\delta_2$). The choice of the uncertainty metric $U$ directly influences the sensitivity and reliability of this criterion.
>
> Specifically, we empirically evaluated three candidate metrics for $U$:
> *   **Negative Maximum Softmax Probability (-MSP):** $U = -\max(\text{Softmax}(\cdot))$. MSP is a standard baseline in OOD detection, favored for its simplicity and computational efficiency. Its dynamics effectively capture the model's transition from uncertainty to certainty during training. A low -MSP value indicates high confidence, directly satisfying the condition $U_i < \delta_1$ and $|\Delta^2 U_i(w^{(t)})| < \delta_2$ for mastered instances.
>
> *   **Entropy:** $U = -\sum p_i \log p_i$. Entropy measures the dispersion of the entire predictive distribution, capturing uncertainty across all classes rather than just the top prediction. A low Entropy value signifies a peaked distribution (high confidence), satisfying the condition $U_i < \delta_1$ and $|\Delta^2 U_i(w^{(t)})| < \delta_2$.
>
> *   **Loss (e.g., Cross-Entropy):** The loss function is the primary driver of optimization. A low loss value indicates that the model's prediction aligns closely with the ground truth, implying high confidence and satisfying the condition $U_i < \delta_1$ and and $|\Delta^2 U_i(w^{(t)})| < \delta_2$.
>
> As shown in Table 8 of the paper, we conducted experiments using these metrics within our UARB framework on the CIFAR-10 benchmark to analyze how the mastery criterion behaves under each metric. The results demonstrate that employing -MSP as the uncertainty score yields the most effective OOD detection performance, achieving the lowest average FPR95 in both near-OOD and far-OOD scenarios. While using Loss or Entropy still provides significant improvements over the baseline, their performance is slightly inferior to -MSP. This empirically confirms that -MSP offers the most reliable and stable signal for our mastery criterion.
>
> > W2. Is the second order difference stable and sufficiently noise-free to reliably identify mastered instances? How does noise in uncertainty estimates affect the criteria?
>
> **Response:** Thank you for this insightful question regarding the stability and noise resilience of the second-order difference in our UARB method.
>
> * **Stability of the Second-Order Difference**. The second-order difference is designed to be stable by capturing the acceleration of uncertainty convergence rather than momentary fluctuations. By evaluating uncertainty changes across three consecutive epochs (t, t-1, t-2), the second-order difference inherently smooths out short-term noise and transient variations in training dynamics. This multi-epoch perspective reduces sensitivity to isolated anomalies or batch-specific noise. The second-order difference emphasizes the rate of change in uncertainty decay. When this value approaches zero, it indicates that uncertainty has stabilized around a plateau, signaling instance mastery. This trend-based approach is less affected by absolute noise levels in individual uncertainty estimates.
>
> * **Noise Resilience of the Second-Order Difference**. Unlike zero-order uncertainty (raw values), the second-order difference acts as a high-pass filter that attenuates low-frequency noise while highlighting consistent convergence patterns. This makes it particularly effective in distinguishing true learning signals from stochastic fluctuations. Our mastery criterion requires both the uncertainty itself and its second-order difference to fall below thresholds ($\delta_1$ and $\delta_2$). This dual requirement ensures that only instances with persistently low and stable uncertainty are classified as mastered, reducing false positives from noisy estimates.

---

> ### Author Response · Authors · 2025-11-24
> **Response to Reviewer LSjt (part2)**
>
> >W3&Q2. It would be great to include sensitivity analyses on $\delta_1$, $\delta_2$ and show robustness of performance.
>
> **Response:** We sincerely thank the reviewer for highlighting the need for sensitivity analyses on $\delta_1$ and $\delta_2$. We add a sensitivity analysis for $\delta_1$ and $\delta_2$ as follows. The results demonstrate that our method is not highly sensitive to the selection of $\delta_1$ and $\delta_2$.
>
>
> |       | Near-Avg (FPR95) | Far-Avg (FPR95) | Near-Avg (AUROC) | Far-Avg (AUROC) |
> |----------------|-----------------|----------------|-----------------|----------------|
> | $\delta_1$=0.01  | 33.10 ± 1.35    | 20.10 ± 0.66   | 89.69 ± 0.49    | 93.54 ± 0.40   |
> | $\delta_1$=0.005 | 33.57 ± 0.46    | 21.39 ± 1.01   | 89.72 ± 0.11    | 92.97 ± 0.44   |
> | $\delta_1$=0.001 | 33.12 ± 1.30    | 21.42 ± 0.56   | 89.71 ± 0.21    | 92.97 ± 0.30   |
> | $\delta_1$=0.05  | 31.71 ± 0.79    | 20.90 ± 1.44   | 90.01 ± 0.24    | 93.30 ± 0.69   |
> | $\delta_1$=0.1   | 32.71 ± 0.51    | 20.81 ± 0.63   | 89.74 ± 0.10    | 93.02 ± 0.34   |
> |
>
> |    | Near-Avg (FPR95) | Far-Avg (FPR95) | Near-Avg (AUROC) | Far-Avg (AUROC) |
> |---------------|----------------|----------------|-----------------|----------------|
> | $\delta_2$=0.001 | 33.10 ± 1.35   | 20.10 ± 0.66   | 89.69 ± 0.49    | 93.54 ± 0.40   |
> | $\delta_2$=0.0005 | 33.69 ± 0.65   | 21.62 ± 0.60   | 89.65 ± 0.24    | 93.06 ± 0.24   |
> | $\delta_2$=0.0001 | 32.47 ± 0.60   | 21.10 ± 0.13   | 89.80 ± 0.28    | 93.19 ± 0.02   |
> | $\delta_2$=0.005 | 33.88 ± 0.93   | 21.45 ± 0.69   | 89.60 ± 0.02    | 92.80 ± 0.39   |
> | $\delta_2$=0.01 | 36.81 ± 0.39   | 21.45 ± 0.24   | 89.41 ± 0.20    | 92.95 ± 0.18   |
> |
>
>
> >W4&Q4. It should report the impact on primary classification task metrics (ID accuracy, calibration, robustness).
>
> **Response:**
> As suggested we report the results as follows.
>
> * **ID accuracy analysis**: We verified the ID classification performance of MSP, MSP+EMA, and their enhanced versions with our method. The experimental results demonstrate that our approach achieves superior ID classification performance.
>
> | Method | ACC |
> |--------|-----|
> | MSP  | 94.96 ± 0.18 |
> | MSP+ours  | 95.03 ± 0.11 |
> | MSP+EMA| 93.80 ± 0.34 |
> | MSP+EMA+ours| 94.50 ± 0.26 |
> |
>
>
> * **calibration analysis**: We evaluated the calibration performance of MSP, MSP+EMA, and their enhanced versions with our method. The experimental results indicate that our approach achieves superior calibration performance.
>
> | Method | ECE |
> |--------|-----|
> | MSP | 2.68 ± 0.12 |
> | MSP+ours | 2.62 ± 0.08 |
> | MSP+EMA | 2.69 ± 0.12 |
> | MSP+EMA+ours| 2.37 ± 0.10 |
> |
>
> * **robustness analysis**: In terms of robustness analysis, we validated the out-of-distribution (OOD) detection performance of the methods under covariate shift scenarios. The results demonstrate that both MSP and MSP+EMA, when enhanced with our method, achieve more robust detection performance.
>
> | Method | FPR95 | AUROC |
> |--------|-----------------|------------------|
> | MSP | 53.29 ± 1.64 | 78.98 ± 0.83 |
> | MSP+ours | 50.37 ± 1.65 |  80.97 ± 0.70 |
> | MSP+EMA | 45.82 ± 0.29 |  81.37 ± 0.14 |
> | MSP+EMA+ours | 44.48 ± 0.71 | 82.12 ± 0.68 |
> |

---

> ### Author Response · Authors · 2025-11-24
> **Response to Reviewer LSjt (part3)**
>
> > W5. It would be great to add the following ablation studies: the effect of using only zero-order vs. also second-order difference; fixed threshold vs adaptive threshold;baseline training without UARB but with same number of epochs/training budget.
>
> **Response:** As suggested we report the ablation results as follows.
> 1. The effect of using only zero-order vs. also second-order difference: We report the results of zero-order, second-order, and zero-order+second-order methods, respectively. The experimental results demonstrate that using the second-order method alone yields better outcomes, and the integration of both methods further enhances the performance of far-OOD detection.
>
> | Method       | Near-Avg (FPR95) | Far-Avg (FPR95) | Near-Avg (AUROC) | Far-Avg (AUROC) |
> |--------------|-----------------|----------------|-----------------|----------------|
> | base (MSP)   | 48.47 ± 0.69    | 31.90 ± 1.51   | 88.21 ± 0.16    | 90.74 ± 0.41   |
> | zero-order            | 37.24 ± 0.46    | 22.06 ± 0.48   | 89.32 ± 0.20    | 92.80 ± 0.04   |
> | second-order            | 32.27 ± 0.80    | 20.83 ± 1.19   | 89.89 ± 0.20    | 93.10 ± 0.33   |
> | zero-order+second-order          | 33.10 ± 1.35    | 20.10 ± 0.66   | 89.69 ± 0.49    | 93.54 ± 0.40   |
> |
>
> 2. Fixed threshold vs adaptive threshold: We conducted experiments comparing fixed threshold and adaptive threshold, and the results (FPR95) demonstrate that the adaptive threshold yields superior performance.
>
> | Method          | MNIST  | SVHN   | Texture  | Places365   |
> |-----------------|----------------|----------------|---------------|---------------|
> | Fixed     | 17.69 ± 3.36   | 13.93 ± 0.42   | 26.60 ± 5.82         | 26.75 ± 1.55        |
> | Adaptive    | 13.15 ± 2.46    | 12.67 ± 3.22   | 23.17 ± 0.46       | 26.24 ± 2.47          |
> |
>
> 3. Baseline training without UARB but with same number of epochs/training budget: We added an ablation study on baseline training without UARB but under the same number of epochs/training budget. The results (FPR95) demonstrate that incorporating UARB into the baseline method leads to a substantial improvement in OOD detection performance, thereby validating the effectiveness of UARB.
>
> | Method        | CIFAR-100     | TIN          | Near-Avg    | MNIST       | SVHN        | Texture     | Places365   | Far-Avg     |
> |---------------|---------------|--------------|-------------|-------------|-------------|-------------|-------------|-------------|
> | baseline      | 52.70 ± 1.47  | 42.13 ± 2.26 | 47.41 ± 0.99 | 25.33 ± 1.50 | 23.77 ± 4.86 | 28.23 ± 0.23 | 40.83 ± 1.74 | 29.54 ± 1.91 |
> | baseline + UARB      | 34.99 ± 0.97  | 29.96 ± 1.03 | 32.47 ± 1.00 | 18.56 ± 2.17 | 13.67 ± 0.86 | 23.42 ± 0.92 | 29.48 ± 1.05 | 21.28 ± 0.75 |
> |
>
> >Q3. It would be great to add experiments on more diverse datasets and OOD types including near-OOD/far-OOD.
>
> **Response:**  To further validate the effectiveness of the method, we also conducted experiments on medical datasets. The in-distribution (ID) dataset is OrganAMNIST, the near-OOD datasets include OrganCMNIST, OrganSMNIST, ChestMNIST, and PneumoniaMNIST, while the far-OOD datasets comprise PathMNIST, DermaMNIST, RetinaMNIST, and BloodMNIST. We report the average FPR95 and AUROC for the corresponding scenarios. The results are shown below. The experimental results demonstrate that the OOD detection performance of the baseline methods improved after integrating our approach, in both near-OOD and far-OOD scenarios, which validates the effectiveness and generalizability of our method.
>
> | Method       | Near-Avg (FPR95) | Far-Avg (FPR95) | Near-Avg (AUROC) | Far-Avg (AUROC) |
> |--------------|---------------|--------------|---------------|--------------|
> | MSP         | 50.38         | 33.91        | 83.08         | 89.23        |
> | MSP+ours     | 42.07         | 25.57        | 86.63         | 91.28        |
> | Energy         | 59.91         | 45.4         | 79.76         | 84.06        |
> | Energy+ours     | 56.28         | 41.57        | 83.75         | 86.31        |
> | KNN          | 35.82         | 14.57        | 86.84         | 93.23        |
> | KNN+ours     | 34.47         | 15.23        | 87.74         | 93.49        |
> | ReAct        | 68.2          | 51.34        | 74.67         | 82.16        |
> | ReAct+ours   | 52.13         | 38.49        | 83.58         | 87.15        |
> | Relation     | 51.49         | 27.25        | 79.53         | 89.45        |
> | Relation+ours| 49.02         | 27.88        | 83.42         | 90.73        |
> | Fdbd         | 47.68         | 22.17        | 82.86         | 91.87        |
> | Fdbd+ours    | 39.35         | 22.18        | 86.06         | 92.54        |
> | Nci          | 72.1          | 46.37        | 69.65         | 81.43        |
> | Nci+ours     | 63.35         | 45.66        | 76.82         | 84.65        |
> |

---

### Author Response · Authors · 2025-12-03
**Rebuttal Summary**

We thank the reviewers for their thorough and constructive comments.
Our key contributions are as follows (from the reviews):

* **On the Core Insight and Problem Motivation:**
Reviewers LSjt and PAyv​ both emphasized the importance of the paper's central observation: that OOD detection performance can degrade in later training epochs due to overfitting. Reviewer PAyv specifically noted that this problem is "clearly demonstrated through experimental evidence," while Reviewer LSjt highlighted that this is an "important" point that is "less emphasized in many OOD works."
* **On the Novelty of the Approach:**
Reviewers LSjt, LZKj, and EbRE​ all praised the innovative angle of the work. Specifically, Reviewer LSjt stated that "intervening in the training loop is a novel angle," and Reviewer LZKj contrasted UARB with "model-centric methods," aligning it with the "growing focus on data quality."
Reviewer EbRE​ elaborated on the originality, noting that UARB addresses overfitting at a "finer, more granular level" than prior methods like UM and "overcomes the limitation of UM’s reliance on pretrained models, enabling direct end-to-end training."
* **On the Methodological Design:**
Reviewers LSjt and LZKj​ commended the core design of the adaptive threshold. They pointed out that by "recognizing that different classes converge at different rates" (LSjt), UARB effectively "balances training of easy and hard classes" (LZKj) through class-specific thresholds.
Reviewers PAyv and LZKj​ appreciated the practical design of UARB. Reviewer PAyv highlighted its strength as a "plug-and-play method with strong applicability," and Reviewer LZKj noted its mechanism of "dynamically filters training data." Reviewer PAyv also found the "motivation behind using uncertainty, the second-order difference, and the adaptive threshold mechanism" to be "clearly explained."
* **On Experimental Validation and Paper Quality:**
Reviewers PAyv and EbRE​ validated the experimental results. Reviewer PAyv stated that "UARB clearly outperforms UM... across multiple benchmarks," and Reviewer EbRE described the experiments as "extensive and comprehensive," demonstrating "promising improvements."
Reviewer EbRE​ also praised the overall quality and clarity of the manuscript, calling it a "good paper with a clear central idea" that is "easy to follow."

Based on the reviewers' valuable feedback, we have conducted a number of additional experiments, which hopefully resolve the reviewers’ concerns. The major additional experiments and improvements are as follows:

* We have added discussion on uncertainty metrics and show how the criteria for mastery behaves under each (LSjt(W1&Q1)).
* We have emphasized the stability and noise resilience of the second-order difference in our UARB method (LSjt(W2)).
* We have included sensitivity analyses on $\delta_1$ and $\delta_2$, and show robustness of performance (LSjt(W3&Q2)).
* We have reported the impact on primary classification task metrics (ID accuracy, calibration, robustness) (LSjt(W4&Q4)).
* We have added the following ablation studies: the effect of using only zero-order vs. also second-order difference; fixed threshold vs adaptive threshold;baseline training without UARB (LSjt(W5)).
* We have added experiments on more diverse datasets and OOD types including near-OOD/far-OOD (LSjt(Q3)).
* We have added a figure in the revisied manuscript that explicitly verify the increase in instances that meet the mastered criteria (PAyv(W1)).
* We have offered computational and memory overhead of UARB (PAyv(W3), LZKj(W4)).
* We have added two new baselines (DICE and Scale) (PAyv(W4)).
* We conducted experiments using other backbone to further assess the applicability of UARB (PAyv(Q1)).
* We have clarified why the first-order difference was not considered (PAyv(Q2)).
* We have conducted an ablation study on $\delta_2$ to analyze sensitivity and impact of second-order difference (PAyv(Q3)).
* We have added Fig 2 in the revised manuscript to directly investigate whether UARB mitigates the overfitting in later training stages. We visualized the t-SNE plots (Fig 3(a) and Fig 3 (b) in the revised manuscript) to demonstrate that our method achieves superior separation (LZKj(W2), EbRE(W3)).
* We have conducted validation on other backbone to further investigate the effectiveness of UARB (LZKj(W3)).
* We have clarified why overfitting to the training data leads to a decline in OOD detection performance (EbRE(W1)).
* We have offered the analysis on whether the hyperparameter settings are sensitive to the dataset (EbRE(W4)).

Although OpenReview experienced a technical issue (on Nov. 27), prior to the outage (on Nov. 24), Reviewer EbRE had acknowledged and affirmed our rebuttal and decided to retain a positive evaluation (score 8: accept).

We thank all reviewers and Area Chairs for you efforts.

Best,
Authors

---

### Meta-Review · Area_Chair_tcH4 · 2025-12-14

**Summary:**

All reviewers acknowledged that focusing on the training stage of OOD detection is motivated. However, reviewers also raised significant concerns especially regarding the evaluations of the proposed method, such as the used architectures, the effectiveness and computational issue of the second-order metrics, etc.  Reviewers also raised concerns about the theoretical analysis of the proposed method.

**Reviewer Concerns:**

Some of the concerns, such as the explanation of used uncertainty metrics, are well addressed. However, the rebuttal is still lacking, in terms of theoretical analysis (only contextual explanations provided), architectures used (the authors blames the effectiveness of ViT model on the problem of OOD detection), and also computational analysis.

**Reviewer Scores:**

The reviewers are less likely to change the scores to positive rating.

---

### Decision · Program_Chairs · 2026-01-26

Reject